# Cisplatin toxicity is counteracted by the activation of the p38/ATF-7 signaling pathway in post-mitotic *C. elegans*

Dorota Raj [1], Bashar Kraish[1], Jari Martikainen [2], Agnieszka Podraza-Farhanieh [1,4], Gautam Kao [1] ✉ & Peter Naredi [1,3] ✉

Cisplatin kills proliferating cells via DNA damage but also has profound effects on post-mitotic cells in tumors, kidneys, and neurons. However, the effects of cisplatin on post-mitotic cells are still poorly understood. Among model systems, *C. elegans* adults are unique in having completely post-mitotic somatic tissues. The p38 MAPK pathway controls ROS detoxification via SKN-1/NRF and immune responses via ATF-7/ATF2. Here, we show that p38 MAPK pathway mutants are sensitive to cisplatin, but while cisplatin exposure increases ROS levels, *skn-1* mutants are resistant. Cisplatin exposure leads to phosphorylation of PMK-1/MAPK and ATF-7 and the IRE-1/TRF-1 signaling module functions upstream of the p38 MAPK pathway to activate signaling. We identify the response proteins whose increased abundance depends on IRE-1/p38 MAPK activity as well as cisplatin exposure. Four of these proteins are necessary for protection from cisplatin toxicity, which is characterized by necrotic death. We conclude that the p38 MAPK pathway-driven proteins are crucial for adult cisplatin resilience.

Cisplatin is a platinum-based chemotherapeutic drug widely used to combat different types of cancer including sarcomas and carcinomas with relatively high treatment efficiency[1,2]. A major mechanism of cisplatin action is via the binding of platinum to DNA and forming intra- and inter-stranded crosslinks. This leads to cell cycle arrest and eventually apoptosis in fast-cycling cells[2]. However, only 1–10% of intracellular cisplatin can be found in nuclei[3,4] and cisplatin has other cellular targets apart from nuclear DNA, such as mitochondria and endoplasmic reticulum[5]. Importantly for our study, many solid tumor cells are post-mitotic[6], and shrinking or eliminating a tumor mass by cisplatin action involves its function in these non-dividing cells as well. Revealing the cellular pathways influenced by cisplatin could provide important information for the design of new cancer treatment strategies targeting slow-cycling or non-dividing cells. Cisplatin therapy is limited by its toxic side effects in auditory neurons and the

kidney[7]. The side effects of toxicity in both neurons and the kidney can be devastating and can even rule out the availability of cisplatin treatment to many individuals. Both organs are completely post-mitotic and the effect of cisplatin needs to be limited or abolished for effective treatment. *C. elegans* is an attractive model for cisplatin resistance studies since somatic tissues of wild-type adult animals are purely post-mitotic and the wild-type worms display resistance to cisplatin treatment[8,9]. The presence of pathway orthologues in this model organism offers the opportunity to use the power of genetics and describe these pathways in vivo.

The innate immune system in *C. elegans* has been extensively studied for its role in protecting worms from bacterial and fungal pathogens[10,11]. For this response, the p38 MAP kinase pathway is one of the three signaling pathways that have the greatest impact on resistance to pathogens[12,13]. In the intestine, the p38 MAP kinase pathway

[1]Department of Surgery, Institute of Clinical Sciences, Sahlgrenska Academy, University of Gothenburg, SE413 45 Gothenburg, Sweden. [2]Bioinformatics and Data Centre, Sahlgrenska Academy, University of Gothenburg, Gothenburg, SE413 45 Gothenburg, Sweden. [3]Department of Surgery, Sahlgrenska University Hospital, SE413 45 Gothenburg, Sweden. [4]Present address: Lundberg Laboratory for Diabetes Research, Department of Molecular and Clinical Medicine, Sahlgrenska Academy, University of Gothenburg, Sweden. ✉e-mail: gautam.kao@gu.se; peter.naredi@gu.se

(NSY-1/MAPKKK, SEK-1/MAPKK, PMK-1/MAPK) acts by activating the transcription factor ATF-7 via phosphorylation in response to pathogen attack[13], whereas for hypodermal pathogens this pathway acts via STA-2 to drive a largely non-overlapping set of genes[14]. Apart from its role in pathogen resistance, the p38 MAPK pathway and the downstream immune response genes are important in the modulation of responses to dietary restriction and aging[15], repair of sterile wounds[16], and the regulation of developmental sleep[17]. Work in *Drosophila* shows that the innate immune genes have roles in processes as diverse as in oogenesis[18] and repair of traumatic brain injury[19]. This has led to the recognition that the immune response genes should be considered as stress modulators or repair factors and that their roles stretch beyond pathogen defense.

Cisplatin induces ROS in mammalian cells[20] and worms[9] and a substantial body of evidence supports the notion that ROS-provoked damage could lead to cell and tissue death. In worms, cisplatin leads to the oxidation of a specific chaperone-type protein ASNA-1. Oxidation of ASNA-1 perturbs the targeting of tail-anchored membrane proteins to the endoplasmic reticulum membrane[9]. However, it is not known whether the levels of ROS species are high enough to inflict damage on tissues at a level sufficient to cause extensive damage or death. ROS detoxification in response to ROS accumulation is driven by the activity of type II detoxification genes which are under control of the SKN-1/NRF transcription factor. Phosphorylation and activation of SKN-1 is in turn mediated upstream by the activation of p38 MAPK pathway components SEK-1 and PMK-1 in response to increased ROS levels[21–23]. It is, therefore, possible to test, using *C. elegans* as a model, whether ROS-induced toxicity is an important aspect of cisplatin-provoked death.

In this study, using a combination of quantitative proteomics, genetic analysis, and selective degradation of signaling molecules we find that the p38 MAP kinase pathway has an important function in ensuring survival on cisplatin. However, we find that the activation of the innate immune response genes downstream of this pathway is far more important than the ROS detoxification arm in mediating the cisplatin response. Restricting the signaling only in somatic tissues of post-mitotic adults is equivalent to a complete knockout of the pathway observed in null mutants. Cisplatin-induced ROS is important for activation of the signaling activity and the transcription factor ATF-7 acting downstream of the p38 MAPK signaling pathway is a central participant in the response. Consistent with this idea we found that PMK-1/MAPK and ATF-7 are phosphorylated upon cisplatin exposure and that ATF-7 phosphorylation requires PMK-1 function. Moreover, IRE-1 functions with TRF-1 in a UPR^ER-independent manner upstream of the p38 MAPK pathway to initiate signaling. We find that sulfenylation of IRE-1 is an early molecular event in response to cisplatin exposure and this molecular landmark in turn leads to the accumulation of innate immune proteins. We identify a set of immune response proteins whose abundance depends on SEK-1 and cisplatin. Strikingly, mutants in several of these innate immune response proteins are also cisplatin sensitive. The mutants accumulate phosphatidyl serine on the surface of the necrotic vacuoles and mutants defective in excretory canal function (the kidney equivalent in worms) function are also cisplatin sensitive. We conclude that in post-mitotic tissues, the IRE-1/PMK-1/ATF-7 pathway function is of greater importance than the ROS detoxification pathway to promote resilience against cytotoxicity.

## Results

### ROS detoxification is not required for survival on cisplatin

There is a well-established function of ROS in the activation of MAPKs when mammalian cells are treated with cisplatin[24]. It was previously shown that cisplatin induces ROS in 1-day-old adult animals[9]. We found evidence that cisplatin treatment leads to ROS-promoted protein damage (Supplementary Fig. 1a, b). Mindful of the role of p38 MAPK activation upon ROS accumulation, we asked if the p38 MAPK cascade

has functions in adult animals in response to cisplatin exposure. We found this to be the case based on several observations. First, cisplatin exposure induced the p38 MAPK pathway activity as monitored by the phosphorylation of PMK-1/MAPK (Fig. 1a, b; Supplementary Fig. 2), and cisplatin-induced ROS was required for p38 activation since treating with a ROS scavenger (MitoTempo) resulted in a decrease in PMK-1 phosphorylation (Fig. 1c, d; Supplementary Fig. 2). Second, cisplatin treatment led to an increase in the accumulation PMK-1::mNeonGreen[25] in the intestinal nuclei in comparison to the untreated controls (Supplementary Fig. 3). Third, mutants in p38 MAPK pathway *sek-1(km4)/MAPKK* and *pmk-1(km25)/MAPK*, were sensitive to cisplatin in comparison to the wild-type animals (Fig. 1e). The $LD_{50}$ of *sek-1(km4)* mutants for cisplatin-induced death was 150 µg/mL (Supplementary Fig. 4), which is half the dose of well-studied cisplatin sensitive mutants in *asna-1*[8] and about 3.5 times lower than that of wild-type animals. Another *sek-1* deletion mutant, *syb2311*, created by precise deletion of the entire coding sequence of *sek-1*, also showed enhanced sensitivity to cisplatin treatment, at levels comparable to those seen in *sek-1(km4)* mutants (Fig. 1e)[25]. We concluded that cisplatin treatment led to activation of the p38 MAPK pathway and that the activation was ROS dependent.

The SKN-1/NRF transcription factor is activated by phosphorylation by PMK-1 in response to elevated ROS levels, and in turn activates the transcription of phase II detoxification genes[23]. We tested the expression of SKN-1-regulated genes *gcs-1*, *gst-4*, *gst-30*, and *gst-38* after cisplatin treatment. Expression of these genes was induced in 1-day adults to a similar extent following a 6 h or 24 h cisplatin exposure (Fig. 1f). The activation of all tested phase II detoxification enzyme genes depended on SKN-1 since there was no induction of these genes upon cisplatin exposure of *skn-1* adult mutants (Fig. 1g). This suggested an important role of SKN-1 dependent activation of phase II detoxification genes upon cisplatin exposure. Surprisingly, adults from three different *skn-1* mutants displayed no sensitivity to cisplatin treatment (Fig. 1h). We concluded that the p38 MAPK pathway plays an important role in cisplatin sensitivity. However, this was not via SKN-1 and the upregulation of the phase II detoxification genes since mutants that failed to upregulate them were not sensitive to cisplatin. DAF-16/FOXO is another key transcription factor in the protection of worms from oxidative stress and *daf-16* mutants are sensitive to pro-oxidants[26]. However, adults harbouring two *daf-16* different null mutants displayed enhanced resistance to cisplatin treatment even at high cisplatin concentrations (Fig. 1h). The lack of sensitivity of *skn-1* and *daf-16* mutants indicated that the ROS levels generated upon exposure were not sufficiently high to cause death if ROS detoxification did not occur via these two major pathways. This finding was in contrast to the importance of DAF-16 and SKN-1 for survival on cisplatin in larval stages which have proliferating somatic tissues[27]. Taken together this analysis showed that the elimination of ROS by detoxifying enzymes was likely not important for the survival of adult worms after cisplatin exposure.

### SEK-1/MAPKK is needed in adult somatic cells for cisplatin resistance

The auxin-inducible degron (AID) system enables the depletion of AID-tagged proteins by tissue-specific TIR1 expression in an auxin-dependent manner[28]. We exposed 1-day-old adult animals expressing the TIR1 pan-somatic driver (*ieSi57*) in *sek-1::cMyc::AID (syb2456)* animals to auxin treatment followed by cisplatin exposure. This allowed us to assess the ability of cisplatin to kill adult worms if SEK-1 is depleted in the soma. Auxin reduced levels of SEK-1::cMyc::AID because a 2 h exposure of 1-day-old adult animals to 1 mM auxin was able to significantly deplete SEK-1, as assessed by decreased levels of phosphorylated/activated PMK-1/p38 (Fig. 2a, Supplementary Fig. 5a). This adult-specific depletion of SEK-1 levels in the soma led to cisplatin sensitivity (Fig. 2b). We failed to directly detect the SEK-1 protein in western blots with the cMyc epitope-tagged allele, and therefore

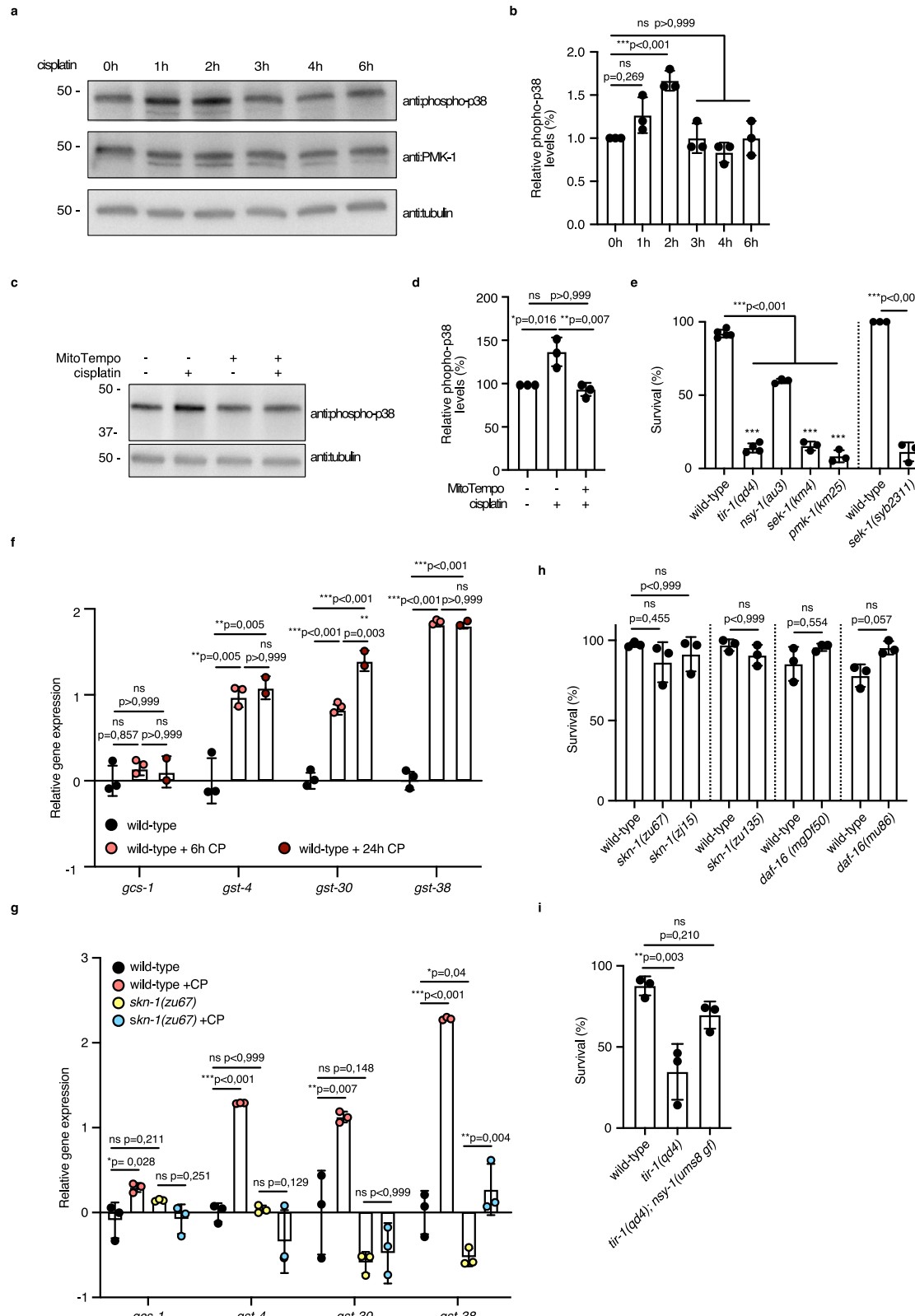

tested a new *sek-1* allele (*syb4220*) in which the 3xFlag::AID tag replaced cMyc::AID. In these animals, expressing the pan-somatic TIR1 driver, the tagged SEK-1 protein could be detected in western blots. Auxin-treated adults had decreased levels of SEK-1 (Fig. 2c, Supplementary Fig. 5b). This auxin treatment regime significantly decreased the survival of 1-day-old adult animals on cisplatin (Fig. 2d). We concluded that the depletion of SEK-1 in post-mitotic tissues was sufficient to induce cisplatin sensitivity and that the proliferating adult germline cells did not contribute significantly to this phenotype.

## Abundance of innate immune response proteins increases after cisplatin treatment

Having ruled out a significant role for ROS as a cause of cisplatin-provoked death in adult animals, we wished to discover pathways and

**Fig. 1 | ROS detoxification does not promote the survival of post-mitotic worms on cisplatin. a** Western blot analysis after SDS-PAGE of phospho-p38 levels in 1-day-old wild-type adults with and without cisplatin treatment for indicated times. Worms were exposed to cisplatin on 300 µg/mL cisplatin-containing plates. The blot was probed with anti:phospho-p38 antibody to detect activated PMK-1, anti:PMK-1 antibody was used to estimate total levels of PMK-1 protein, tubulin was used as a loading control. Full uncropped images are available in Supplementary Fig. 2. **b** Relative phospho-p38 level quantification from blots presented in Supplementary Fig. 2. Statistical significance was determined by one-way ANOVA followed by Bonferroni post hoc correction. The experiment was performed in triplicate. Bars represent mean ± SD. **c** Western blot analysis after SDS-PAGE of phospho-p38 levels in 1-day-old wild-type adults with (+) and without (−) Mito-Tempo/cisplatin exposure. The blot was probed with anti:phospho-p38 antibody, tubulin was used as a loading control. Full uncropped images are available in Supplementary Fig. 2. **d** Relative phospho-p38 levels quantification from blots

presented in Supplementary Fig. 2. Statistical significance was determined by one-way ANOVA followed by Bonferroni post hoc correction. The experiment was performed in triplicate. Bars represent mean ± SD. **e, h, i** Mean survival ± SD of 1-day-old adults with the indicated genotypes after 24 h cisplatin (300 µg/mL) exposure. Statistical significance determined was by one-way ANOVA followed by Bonferroni post hoc correction. The experiment was performed in triplicate. All data on cisplatin sensitivity trials is available in Supplementary Data 4. **f, g** Relative gene expression of four phase II detoxification genes in 1-day-old adult wild-type animals (**f**) without or with (+CP) 6 h or 24 h cisplatin treatment, **g** without or with (+CP) 6 h cisplatin treatment. Worms were exposed to cisplatin on 300 µg/mL cisplatin-containing plates. Statistical significance was determined by one-way ANOVA followed by Bonferroni post hoc correction. The experiment was performed in triplicate for 6 h exposure timepoint and duplicate for 24 h exposure timepoint. F44B9.5 was used as a normalizing control. Bars represent mean ± SEM. Source data are provided as a Source Data file.

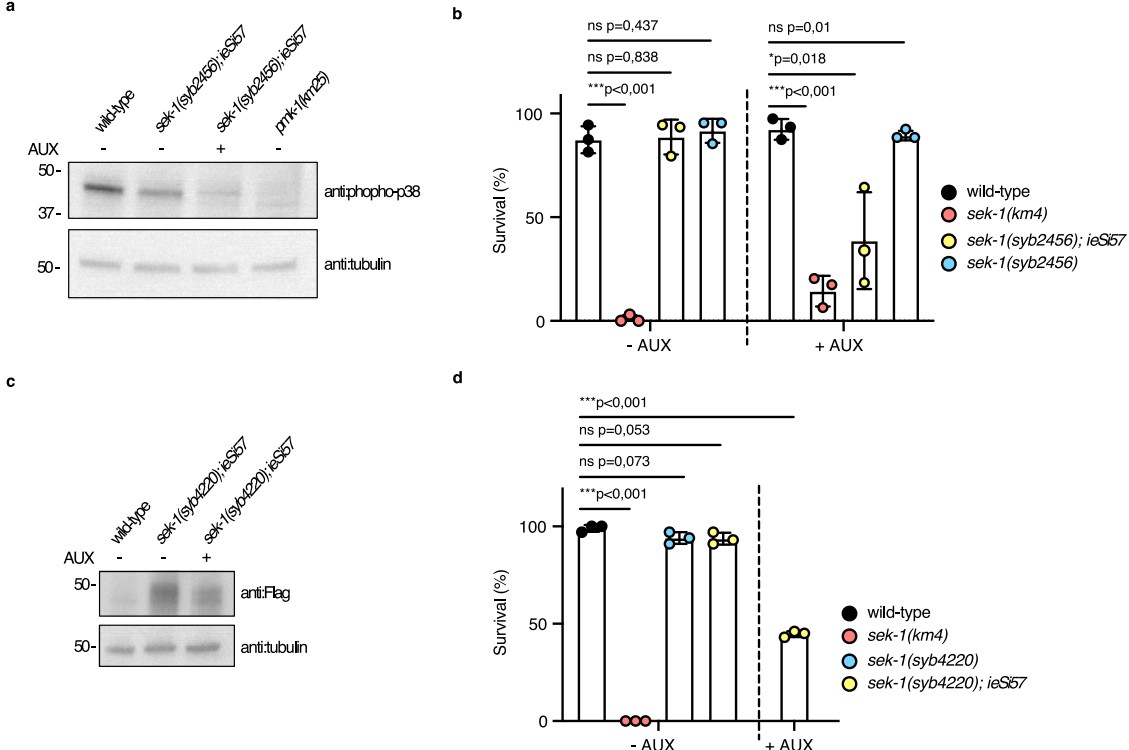

**Fig. 2 | SEK-1 is needed in adult somatic cells for cisplatin resistance. a** Western blot analysis after SDS-PAGE of phospho-p38 levels in 1-day-old adults with (+) and without (−) 1 mM auxin (AUX) treatment. The blot was probed with anti:phospho-p38 antibody and tubulin was used as a loading control. Full uncropped images are available in Supplementary Fig. 5. **b, d** Mean survival ± SD of 1-day-old adults with the indicated genotypes after 24 h exposure to 300 µg/mL cisplatin-containing plates without (−AUX) or with (+AUX) 1 mM auxin treatment (see the "Methods"

section for details). Statistical significance was determined by the **b** independent two-sided *t*-test or **d** one-way ANOVA followed by Bonferroni post hoc correction. The experiment was performed in triplicate. **c** Western blot analysis after SDS-PAGE of SEK-1 levels in 1-day-old adults with (+) and without (−) 1 mM auxin (AUX) treatment. The blot was probed with anti:Flag antibody and tubulin was used as a loading control. Source data are provided as a Source Data file.

biological processes downstream of p38 MAPK signaling that may be involved in cisplatin resistance. To this end, we performed quantitative proteomics analysis to identify proteins that changed in abundance in response to cisplatin exposure (Fig. 3a). Principal component analysis (PCA) showed distinct abundance profiles in cisplatin-treated and control samples and demonstrated the consistent reproducibility between the biological replicates (Supplementary Fig. 6). Overall, 3586 proteins had significant changes in abundance (FDR < 0.05) (Supplementary Data 1). Of these, 121 proteins had a fold change increase >2, and 158 proteins were decreased in abundance by <−2 fold change (Fig. 3b), demonstrating a robust response to cisplatin.

To further explore which proteins might account for the response to cisplatin treatment, we performed Reactome enrichment analysis on differentially regulated proteins (FDR < 0.05). The proteins fell into

multiple functional categories (Supplementary Data 2). The highest count (166 proteins) belonged to members of the innate immune system (Fig. 3c), followed by vesicle-mediated transport and membrane trafficking (Fig. 3c). In addition, this set of proteins (FDR < 0.05 and FC < −2 or FC > 2) were found to be involved in diverse biological functions; for example, response to heat, IRE1-mediated unfolded protein response, and innate immune response (Supplementary Data 2).

## p38 MAPK-driven expression of immune genes is induced by cisplatin

The p38 MAP kinase signaling pathway is essential for *C. elegans* immune response and anti-microbial defense[12,13,29,30]. In addition to previously characterized *sek-1* and *pmk-1* mutants as sensitivity factors

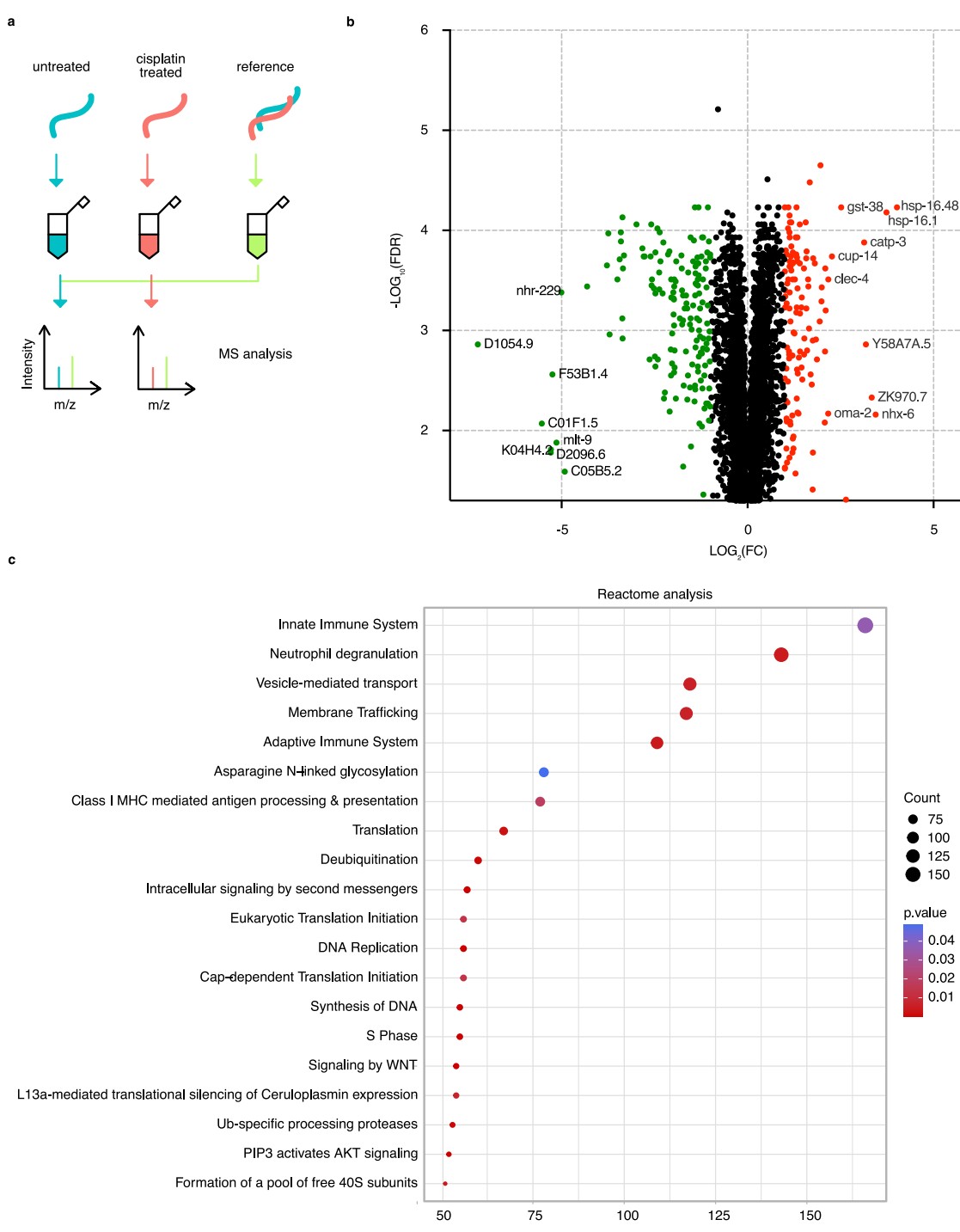

**Fig. 3 | The abundance of innate immune response proteins increases after cisplatin treatment. a** Overview of the proteomics experimental setup. Cisplatin exposure was 300 µg/mL for 6 h (see Materials and Methods section for details). **b** Volcano plot of differentially cisplatin-induced proteins (FDR < 0.05). Proteins that had a positive fold change (FC > LOG$_2$(2)) upon cisplatin treatment are indicated in red and negative fold change (FC < -LOG$_2$(2)) upon cisplatin treatment in green. FC, fold change. FDR, false discovery rate. For details see Supplementary Data 2. **c** Reactome analysis with overrepresented pathways (FDR < 0.05). For details see Supplementary Data 1 and Supplementary Data 2. For Reactome enrichment analysis the hypergeometric model was implement to assess whether the number of selected genes associated with Reactome pathway is larger than expected. The p values were calculated based the hypergeometric model.

(Fig. 1a), we tested loss-of-function mutants in the upstream components, *nsy-1/MAPKKK*, and *tir-1/SARM1*, and found these mutants also to be cisplatin sensitive (Fig. 1a). Notably, *tir-1(qd4)* mutants have only been shown to have defects in innate immune function[31] indicating the likely involvement of the innate immune system in this response[15]. The cisplatin sensitivity phenotype of *tir-1(qd4)* mutants was suppressed by *nsy-1(ums8 gf)* gain-of-function mutants[30] (Fig. 1i) indicating that the

canonical signaling pathway was operating. PMK-1-mediated signaling controls the expression of numerous innate immune response genes[12,13,30]. Since *pmk-1* mutants were cisplatin sensitive, we next investigated whether the expression of PMK-1-dependent immune response genes was affected by cisplatin exposure. qRT-PCR analysis indicated that many of these genes were significantly upregulated (Fig. 4a), supporting the notion that the immune response participates

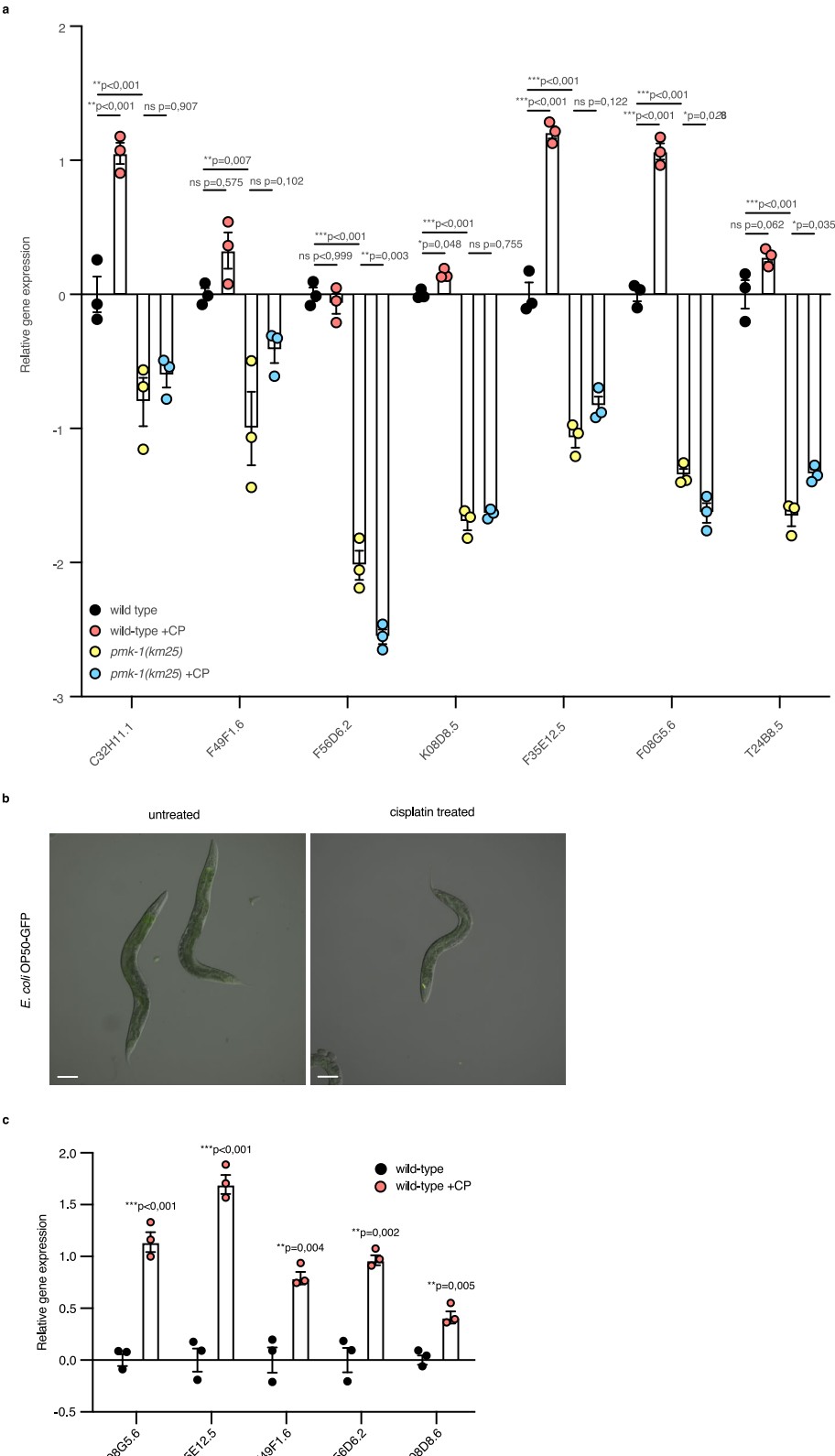

in or responds to cisplatin treatment. PMK-1 was required for cisplatin-dependent induction of these immune genes since all these genes required *pmk-1* for their cisplatin-provoked upregulation (Fig. 4a), as well as for their basal expression (Fig. 4a), as reported before in the context of bacterial infection response[12]. We concluded that the entire signal transduction pathway (TIR-1, NSY-1, SEK-1 and PMK-1) was involved in mediating the response to the cisplatin challenge.

*E. coli* accumulating in the intestine can provoke the worm's innate immune response and lead to phenotypic consequences[32]. We asked whether the accumulation of bacteria in cisplatin-treated animals was the trigger for the induction of the immune response. We did not detect any GFP-labeled bacteria in the intestinal cells of drug-exposed worms or any damage to the intestinal cells in the time period when immune response genes are induced in the intestine

**Fig. 4 | p38 MAPK-driven immune genes expression is induced by cisplatin and is independent of the presence of the bacteria. a** Relative gene expression of PMK-1-dependent immune genes in 1-day-old adult animals with the indicated genotypes without or with (+CP) 6 h cisplatin treatment. Worms were exposed to cisplatin on 300 μg/mL cisplatin-containing plates. Statistical significance was determined by one-way ANOVA followed by Bonferroni post hoc correction. The experiment was performed in triplicate. F44B9.5 was used as a normalizing control. Bars represent mean ± SEM. **b** Representative bright-field photographs of *E. coli* OP50-GFP throughout the intestine of wild-type animals without or with 6 h cisplatin treatment (*n* = 10). Worms were exposed to cisplatin on 300 μg/mL cisplatin-containing plates. Scale bar, 100 μm. **c** Relative gene expression of PMK-1-dependent immune genes in 1-day-old wild-type adult animals in the absence of OP50 without or with (+CP) 3 h cisplatin treatment. Worms were exposed to 300 μg/mL cisplatin in liquid. Statistical significance was determined by independent the two-sided *t*-test. The experiment was performed in triplicate. F44B9.5 was used as a normalizing control. Bars represent mean ± SEM. Source data are provided as a Source Data file.

(Fig. 4b). *E. coli* did not accumulate in the intestinal lumen in drug-exposed worms, a condition that can induce the immune response. Furthermore, *E. coli* were killed on the agar plates containing cisplatin within 6 h, which would prevent them from inducing an immune response. Moreover, PMK-1 phosphorylation as well as expression of known p38-dependent immune response genes were induced by cisplatin exposure even in the absence of bacteria (Fig. 4c). We concluded that the p38 MAPK pathway drives immune response gene expression as an early response directly to cisplatin challenge and not via an effect produced by bacteria.

## The transcription factor ATF-7 is required for cisplatin resistance

We further tested the notion that the p38 MAPK pathway activates the innate immune pathway in response to cisplatin by assessing the role of ATF-7. ATF-7 is an ortholog of the mammalian ATF2 family of basic region leucine zipper (bZIP) transcription factors and a key downstream target of the p38/PMK-1 phosphorylation in *C. elegans*[13]. ATF-7 has a crucial role as a transcriptional regulator of the PMK-1-mediated intestinal innate immune response[13]. *atf-7(qd22)* mutants display diminished expression of PMK-1-dependent immune genes resulting in a strong pathogen susceptibility phenotype[13]. *atf-7(qd22)* mutants were sensitive to cisplatin treatment and the phenotype was further enhanced by *pmk-1* loss-of-function (Fig. 5a), indicating the possibility of cross-talk between PMK-1 and other signaling pathways or that PMK-1 modulates the activation of other transcription factors in response to cisplatin challenge. qRT-PCR analysis showed that the induction of the *pmk-1* dependent immune genes after cisplatin exposure was abrogated in *atf-7(qd22)* mutants (Supplementary Fig 7). *atf-7(qd22 qd130)* mutant worms allow partial expression of immune genes[31]. These mutants were cisplatin-resistant (Fig. 5b) and the *atf-7(qd22qd130)* mutation suppressed the *sek-1* and *pmk-1* mutant cisplatin sensitivity phenotype (Fig. 5b) indicating its role downstream of the MAPK cascade in the context of cisplatin response. We concluded that increased gene expression in *atf-7(qd22 qd130)* mutants likely ensures survival upon cisplatin exposure and was sufficient for the suppression of the *sek-1* and *pmk-1* cisplatin sensitivity phenotype. The transcriptional GFP reporter for the p38-ATF-7 regulated immune response gene, T24B8.5 expressed from the *agIs219* transgene, is a reporter of p38/MAPK pathway activity[31]. Cisplatin treatment increased the T24B8.5p::GFP levels in intestinal cells (Fig. 5c) and this activation was *pmk-1* and *sek-1* dependent (Fig. 5c) indicating that the intestine was likely an important site for cisplatin action.

## The epidermal p38 MAPK pathway is not involved in the cisplatin response

The p38 MAP kinase pathway promotes innate immunity both in the intestine and in the epidermis (hypodermis) via modulation of different transcription factors and distinct sets of immune effectors[14,33]. The genetic analysis of *atf-7* alleles (Fig. 5a, b), increased levels of ATF-7::EGFP in intestinal nuclei (Supplementary Fig. 8a, b), and induction of the intestinal T24B8.5p::GFP reporter (Fig. 5c) indicated that the intestinal immunity pathway driven by p38/MAPK was an important participant in cisplatin response. We asked whether specifically perturbing the epidermal arm of the innate immune response would also affect survival on cisplatin. We, therefore, examined mutants in three genes participating

in epidermal immunity; damage-associated molecular pattern (DAMP) receptor *dcar-1*, a component of the signaling cascade *gpa-12*, and the transcription factor *sta-2*, which is phosphorylated and activated by the p38 MAPK cascade. All three mutants were as resistant as wild-type animals to cisplatin (Supplementary Fig. 9a, b). To further support our notion that hypodermal tissue was not important for this response, we used the auxin-inducible degron (AID) system to assess if hypodermal SEK-1 depletion affected cisplatin survival. We exposed 1-day-old adult animals expressing the TIR1 hypodermal driver (*reSi1*)[34] in the *sek-1::cMyc::AID (syb2456)* background to auxin treatment followed by cisplatin exposure. Growing these animals from the L1 larval stage to adulthood in the presence of auxin did not result in a cisplatin sensitivity phenotype (Supplementary Fig. 9c). By contrast, this treatment condition significantly decreased cisplatin survival of worms expressing the TIR1 intestinal driver (*ieSi61*) in *sek-1::cMyc::AID (syb2456)* background (Supplementary Fig. 9c). These experiments supported our conclusion that activation of the p38 MAPK pathway in the intestine is necessary for protection from cisplatin whereas the epidermal arm of the p38 MAPK signaling cascade did not play a significant role in cisplatin response.

## ATF-7 phosphorylation occurs in response to cisplatin exposure

ATF-7 is phosphorylated by the action of SEK-1 and PMK-1[13]. To further explore molecular mechanisms that promote cisplatin-dependent immune gene expression, we directly probed the phosphorylation state of ATF-7 after cisplatin exposure. We monitored the phosphorylation status of ATF-7 using Phos-tag gels in which phosphorylated protein species migrated more slowly than the non-phosphorylated forms. We found that levels of phosphorylated ATF-7::TY1::EGFP::3x-Flag expressed from the *wgIs638* transgene[35] were higher after cisplatin treatment compared to untreated worms (Fig. 6a, b; Supplementary Fig. 10). The specificity of the phospho-ATF-7 band was confirmed by Lambda Protein Phosphatase (Lambda PP) treatment, which dephosphorylates phosphorylated serine, threonine, and tyrosine residues. With Lambda PP treatment, no phospho-ATF-7 band was observed (Fig. 6a). Phospho-ATF-7 was not detected in *pmk-1(km25)* mutants with or without cisplatin treatment (Fig. 6c, Supplementary Fig. 10). This confirmed the identification of the phosphorylation status of the shifted species and is consistent with the finding that PMK-1 phosphorylates and activates ATF-7[13]. Furthermore, standard SDS-PAGE showed that increased steady-state levels of AFT-7::TY1::EGFP::3xFlag were detected after exposure of worms to cisplatin for 6 h (Fig. 6d, e; Supplementary Fig. 10). We found that the total levels of the tagged ATF-7 protein were stable in *pmk-1(km25)* mutants (Fig. 6d, e), and *atf-7* transcript levels were comparable to those in *pmk-1(+)* worms (Fig. 6f). Fluorescence microscopy of *wgIs638* expressing animals showed that ATF-7::TY1::EGFP::3xFlag levels were increased significantly after cisplatin exposure in the nuclei of intestinal cells (Supplementary Fig. 8a, b). The tagged ATF-7 protein produced by the *wgIs638* transgene is functional since it could rescue the T24B8.5p::GFP expression block in *atf-7(qd22)* mutants (Supplementary Fig. 8c).

## IRE-1 functions upstream of the p38 MAPK pathway in cisplatin response

It has been previously shown that ROS exposure leads to sulfenylation of IRE-1 on the cysteine 663 residue. This not only inhibits the

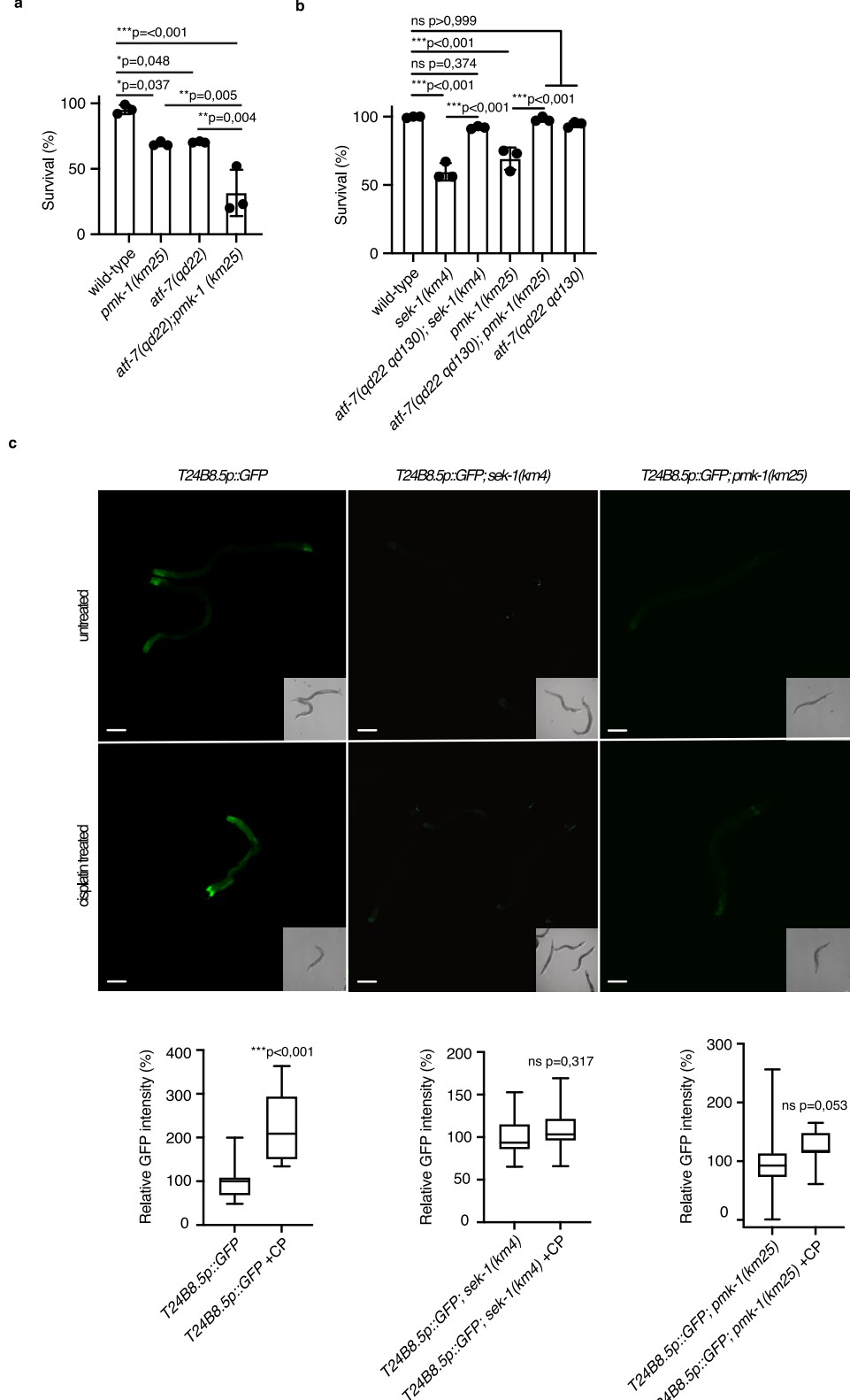

**Fig. 5 | The ATF-7 transcription factor is required for cisplatin resistance in C. elegans. a**, **b** Mean survival ± SD of 1-day-old adults with the indicated genotypes after 24 h cisplatin (150 µg/mL) exposure. Statistical significance was determined by one-way ANOVA followed by Bonferroni post hoc correction. The experiment was performed in triplicate. **c** Expression from the T24B8.5p::GFP reporter imaged by fluorescence microscopy in the 1-day-old adult wild-type, *sek-1(km4)*, and *pmk-1(km25)* animals without or with cisplatin treatment. Worms were exposed to cisplatin for 6 h on plates containing 300 µg/mL cisplatin. Relative T24B8.5p::GFP intensity quantification in the wild-type, *sek-1(km4)*, and *pmk-1(km25)* animals background without or with (+CP) cisplatin treatment. Box plots indicate median (middle line), 25th, 75th percentile (box) and Min to Max (whiskers). Statistical significance was determined by the independent two-sided t-test for wild-type and *sek-1(km4)* animals or Mann–Whitney U Test for *pmk-1(km25)* animals. Source data are provided as a Source Data file.

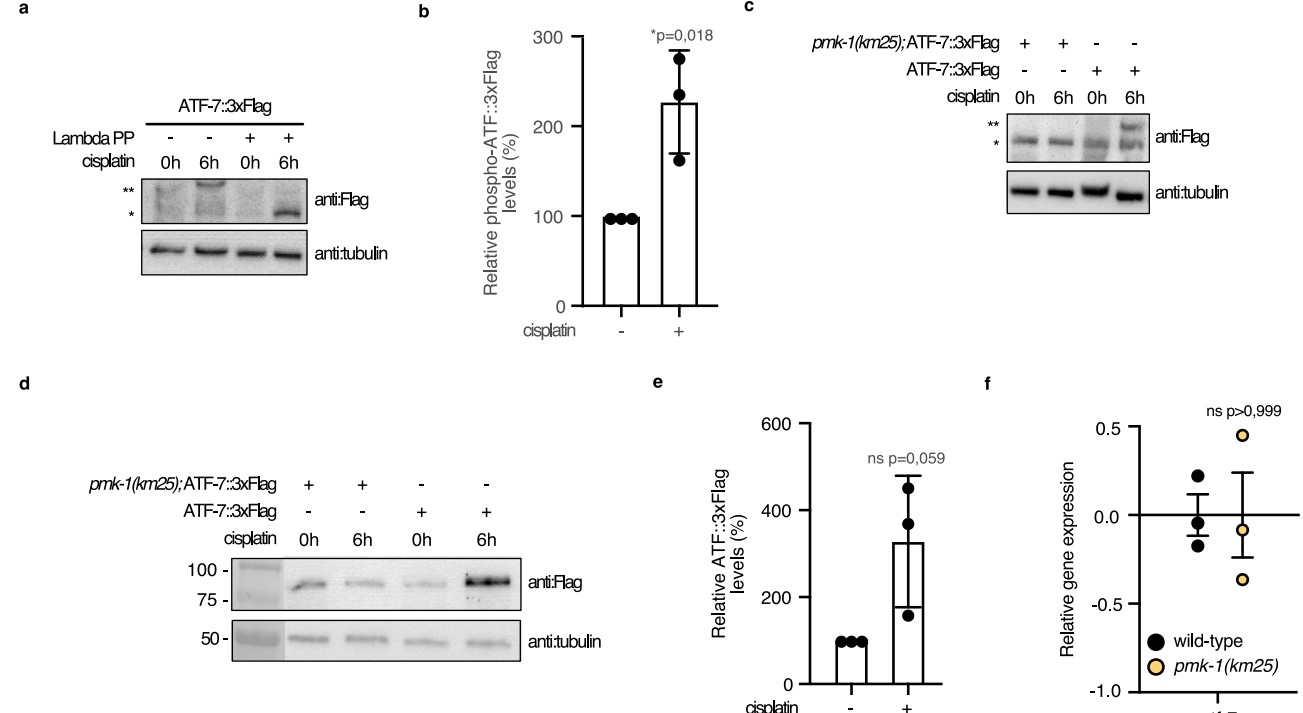

**Fig. 6 | ATF-7 phosphorylation occurs in response to cisplatin exposure.**
**a**, **c** Phosphorylation level analysis of ATF-7 in 1-day-old adult wild-type animals expressing ATF-7::TY1::EGFP::3xFLAG from the *wgIs638* transgene, and 1-day-old adult animals expressing *wgIs638* transgene in *pmk-1(km25)* mutants, without or with cisplatin treatment for indicated times as well as without or with Lambda Protein Phosphatase (Lambda PP) treatment. Worms were exposed to cisplatin on 300 µg/mL cisplatin-containing plates. * non-phosphorylated protein, ** phosphorylated protein. The blots were probed with anti:Flag antibody and tubulin was used as a loading control. **b** Relative phospho-ATF-7::3xFlag levels quantification from blots presented in Supplementary Fig. 10. Statistical significance was determined by an independent two-sided t-test. The experiment was performed in triplicate. Bars represent mean ± SD. **d** Western blot analysis after SDS-PAGE of ATF-7 total protein levels in 1-day-old adult wild-type animals, 1-day-old adult animals

expressing ATF-7::TY1::EGFP::3xFLAG from the *wgIs638* transgene, and 1-day-old adult animals expressing *wgIs638* transgene in *pmk-1(km25)* mutants, without or with cisplatin treatment for indicated times. Worms were exposed to cisplatin on plates containing 300 µg/mL cisplatin. The blots were probed with anti:Flag antibody and tubulin was used as a loading control. **e** Relative ATF-7::3xFlag levels quantification from blots presented in Supplementary Fig. 10. Statistical significance was determined by an independent two-sided t-test. The experiment was performed in triplicate. Bars represent mean ± SD. **f** Relative *atf-7* gene expression in 1-day-old adult wild-type and *pmk-1(km25)* animals. Statistical significance was determined by the independent two-sided t-test. The experiment was performed in biological triplicates. F44B9.5 was used as a normalizing control. Bars represent mean ± SEM. Source data are provided as a Source Data file.

canonical UPR^ER activity of IRE-1 but promotes a redox-regulated role of IRE-1 in p38 MAPK pathway activation via the TRF-1 protein[36]. Therefore, we tested whether IRE-1 was the upstream signaling component for activation of the p38 MAPK pathway upon cisplatin treatment. First, we asked if the cisplatin exposure would lead to the induction of UPR^ER. No induction of the UPR^ER marker *hsp-4* mRNA was observed after cisplatin treatment (Fig. 7a). Second, we asked if the knockdown of IRE-1 would cause cisplatin sensitivity. Indeed, two *ire-1* mutants showed enhanced cisplatin sensitivity (Fig. 7b). However, *xbp-1* mutants were not cisplatin sensitive (Fig. 7c) indicating that IRE-1 did not act through the canonical UPR^ER signaling pathway. Third, using a biotin-linked Dimedone derivative (DCP-BIO1), we determined that IRE-1::3xFlag (from the *syb5891* allele) showed significantly increased sulfenylation in vivo in response to cisplatin exposure (Fig. 7d, e; Supplementary Fig. 11). Moreover, cisplatin-induced rapid sulfenylation of IRE-1::3xFlag, but not of IRE-1^C663S::3xFlag (from the *syb6493-syb5891* allele), indicated that IRE-1 sulfenylation in response to cisplatin occurs via the previously identified C663 residue (Fig. 7d, e; Supplementary Fig. 11). IRE-1 acts via the *trf-1/TRAF2* gene to activate NSY-1/MAPKKK in response to ROS[36]. We found that *trf-1(nr2014)* mutants were also cisplatin sensitive (Fig. 7b). The cisplatin sensitivity phenotype of *ire-1* mutants was suppressed by a gain-of-function mutation in *tir-1*[37] (Fig. 7f) and the sensitivity of *trf-1* mutants was suppressed by *nsy-1(ums8 gf)* (Fig. 7g), showing that the IRE-1-TRF-1 module acted upstream of the p38 MAPK pathway consistent with

earlier findings. Further, we found that the upregulation of the p38-ATF-7 regulated reporter T24B8.5p::GFP upon cisplatin exposure was suppressed in *ire-1* mutants (Fig. 7h). These lines of evidence led us to propose, that TRF-1 might allow recruitment of NSY-1 to IRE-1 which in turn will lead to p38 MAPK activation as has been reported for acute ROS activation[36]. Further, the data supported the idea that cisplatin provoked an increase in ROS levels leading to the sulfenylation of IRE-1 and the consequent downstream activation of the p38 MAPK pathway.

## Identification of effectors modulated by cisplatin and SEK-1

We found that the innate immunity pathway was the top cisplatin-enriched response (Fig. 3c). and that the innate immune response driven via the p38 MAPK/ATF-7 pathway was important for protection from cisplatin cytotoxicity. A well-established set of PMK-1-dependent genes were reporters for pathway activity (Fig. 4a). We wished to determine next whether the immune effector proteins were just reporters for cisplatin exposure, or had a functional role in the process of establishing resilience against cisplatin cytotoxicity. We therefore sought to identify candidate p38 MAPK-dependent genes that might play a role in resilience. With this in mind, we performed a second proteomics analysis in *sek-1(km4)* mutants in order to characterize the SEK-1-dependent immune response and to identify genes that might play an important role downstream of SEK-1 (Fig. 8a, Supplementary Data 1). Principal component analysis (PCA) showed distinct expression profiles between *sek-1(km4)* and

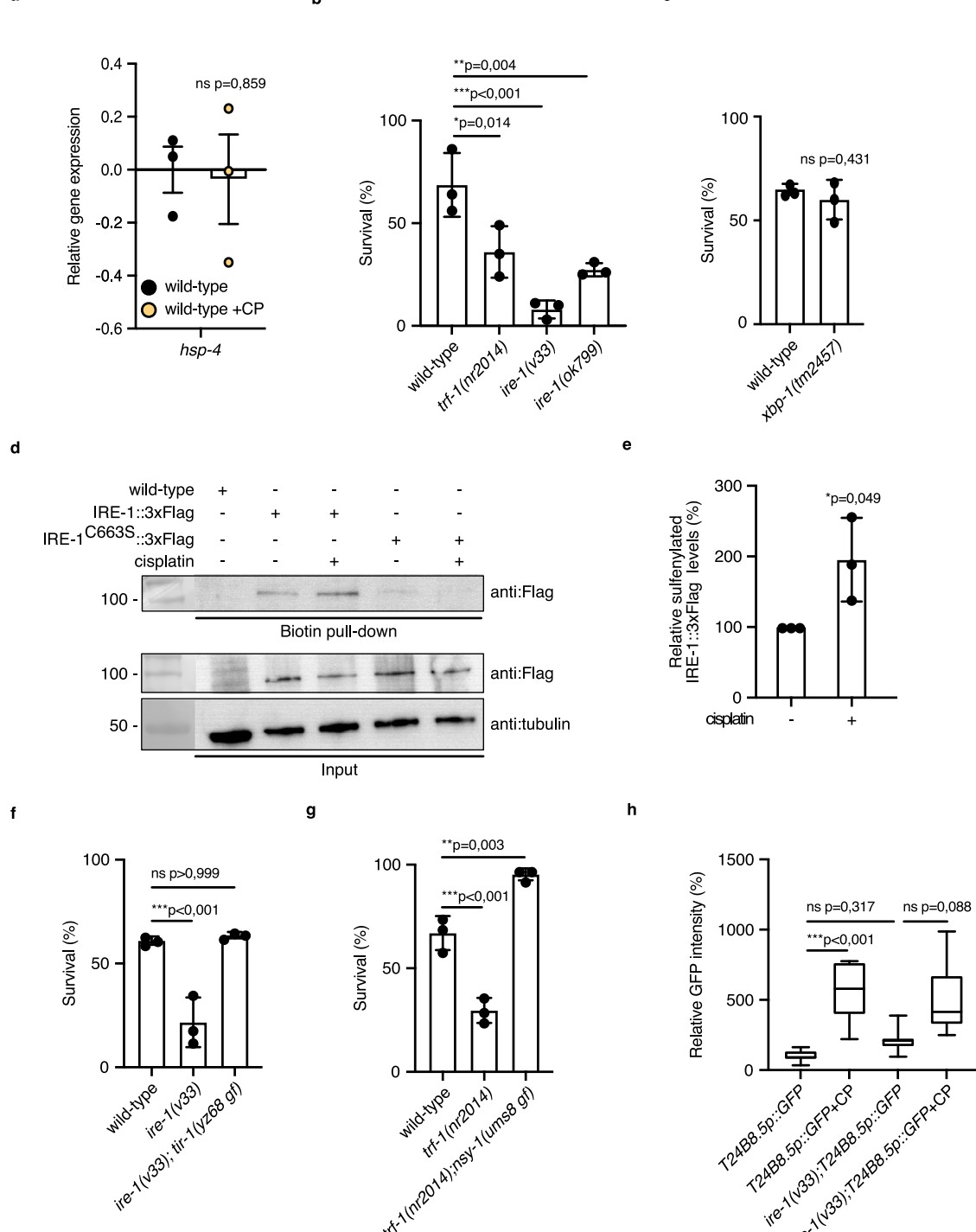

**Fig. 7 | The IRE-1/TRF-1 module functions upstream of the p38 MAPK pathway to activate cisplatin response. a** Relative *hsp-4* gene expression in 1-day-old adult wild-type animals without or with (+CP) 6 h cisplatin treatment. Worms were exposed to cisplatin on - plates containing 300 μg/mL cisplatin. Statistical significance was determined by the independent two-sided *t*-test. The experiment was performed in triplicate. F44B9.5 was used as a normalizing control. Bars represent mean ± SEM. **b, c, f, g** Mean survival ± SD of 1-day-old adults with the indicated genotypes after 24 h cisplatin (450 μg/mL) exposure. Statistical significance was determined by the **c** independent two-sided *t*-test or **b, f, g** one-way ANOVA followed by Bonferroni post hoc correction. The experiment was performed in triplicate. **d** Sulfenylation of IRE-1 using wild-type animals, animals expressing IRE-1::3xFlag or IRE-1^C663S^::3xFlag without (−) or with (+) cisplatin treatment (300 μg/mL

of cisplatin for 2 h) The blots were probed with anti:Flag antibody and tubulin was used as a loading control. **e** Relative sulfenylated IRE-1::3xFlag levels quantification from blots presented in Supplementary Fig. 11. Statistical significance was determined by an independent two-sided t-test. The experiment was performed in triplicate. Bars represent mean ± SD. **h** Relative T24B8.5p::GFP intensity quantification in the wild-type and *ire-1(v33)* animals background without (WT: $n = 13$; *ire-1(v33)*: $n = 7$) or with (+CP) cisplatin treatment (WT: $n = 12$; *ire-1(v33)*: $n = 13$). Worms were exposed to cisplatin for 6 h on 300 μg/mL cisplatin-containing plates. Box plots indicate median (middle line), 25th, 75th percentile (box) and Min to Max (whiskers). Statistical significance was determined by the Kruskal–Wallis test followed by Dunn's post hoc correction. Source data are provided as a Source Data file.

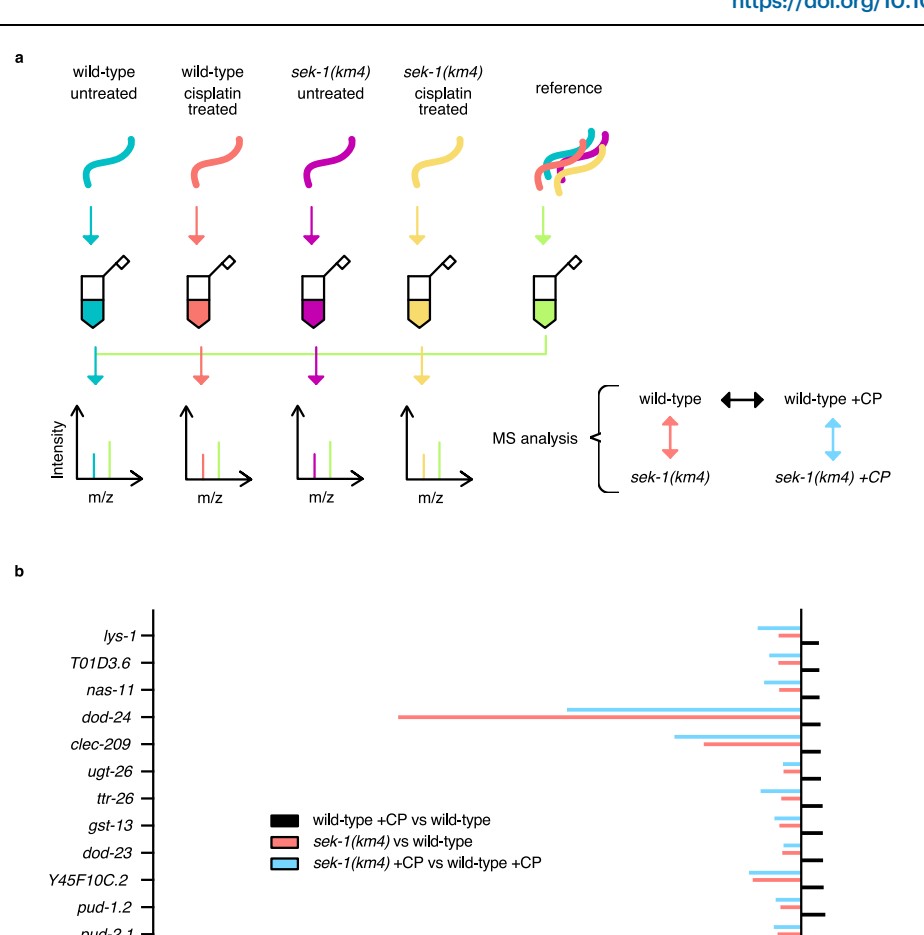

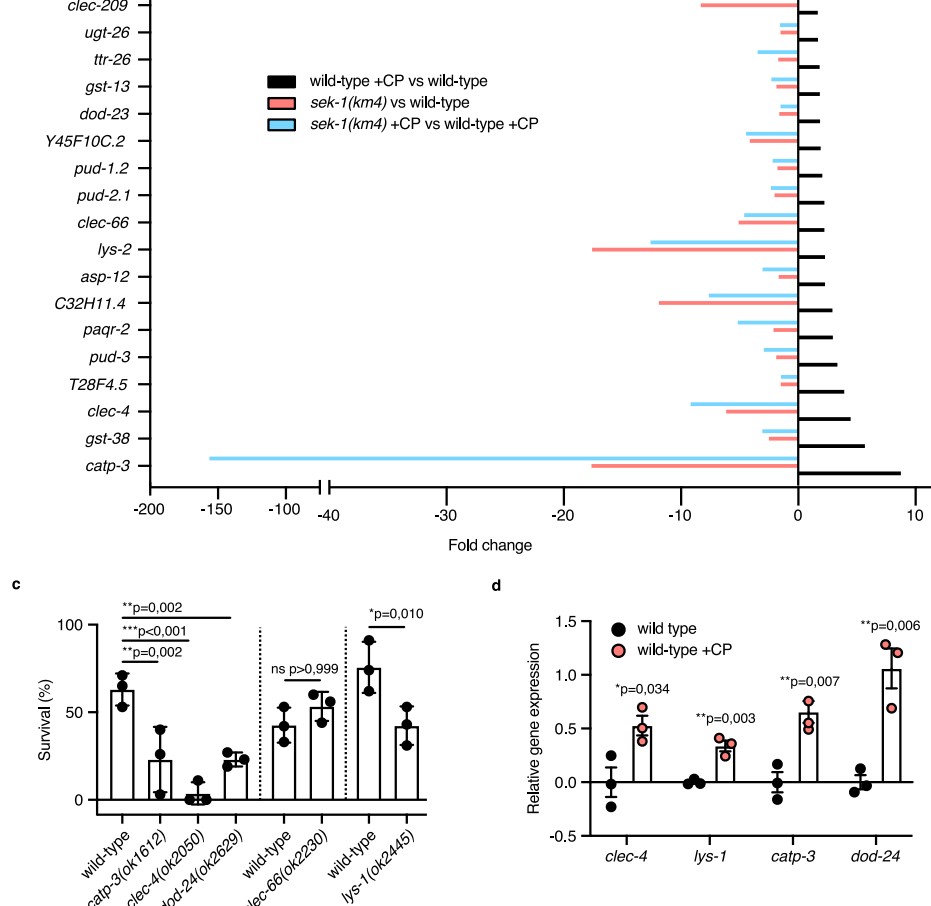

**Fig. 8 | Identification of immune effectors whose abundance is affected by cisplatin and SEK-1. a** Overview of the proteomics experimental setup. **b** Protein fold change (FDR < 0.05) comparison in three different experimental setups. Cisplatin exposure: 300 µg/mL for 6 h. See the Result section for details. **c** Mean survival ± SD of 1-day-old adults with the indicated genotypes after 24 h cisplatin (450 µg/mL) exposure. Statistical significance was determined by one-way ANOVA followed by Bonferroni post hoc correction. The experiment was performed in triplicate. **d** Relative gene expression of SEK-1-dependent immune genes in 1-day-old adult animals with the indicated genotypes without or with (+CP) cisplatin treatment. Worms were exposed for 6 h to cisplatin on 300 µg/mL cisplatin-containing plates. Statistical significance was determined by an independent two-sided *t*-test. The experiment was performed in triplicate. F44B9.5 was used as a normalizing control. Bars represent mean ± SEM. Source data are provided as a Source Data file.

wild-type samples (Supplementary Fig. 12a) as well as between *sek-1(km4)* and wild-type samples treated with cisplatin (Supplementary Fig. 13a). 1307 proteins displayed significantly increased abundance (FDR < 0.05) (Supplementary Data 1, Supplementary Fig. 12b) and 129 proteins showed decreased abundance with a fold change less than −1.5 in *sek-1* mutants (Supplementary Fig. 12b). Proteins that participate in the immune response were a prominent class among these. Cisplatin treatment changed the abundance of 2729 proteins in *sek-1(km4)* samples in comparison to wild-type samples (FDR < 0.05) (Supplementary Data 1, Supplementary Fig. 13b), and again proteins that participate in the immune response were a prominent class among the most downregulated proteins (FC < −1.5) (Supplementary Fig. 13b). Based on these results we anticipated that SEK-1-dependent proteins characterized in the second proteomics profiling were likely important for protection from cisplatin toxicity. We discovered that of 325 proteins (FDR < 0.05) with an FC > 1.5 upon cisplatin treatment in the wild-type background, 22 of them were downregulated with FC < −1.5 in *sek-1* mutants background (in comparison to the wild-type samples), and had at least 1.5-fold decrease after cisplatin treatment in *sek-1* mutants background in comparison to cisplatin-treated wild-type animals (Fig. 8b, Supplementary Data 3). The regulation of these proteins was not only highly p38 pathway-dependent but their upregulation might be crucial for the survival of post-mitotic cells upon cisplatin treatment. Many of the immune genes present in the SEK-1-dependent set are known p38 MAPK target genes[12]. Taken together, we identified the PMK-1/p38 MAPK signaling pathway as a central component in protection against cisplatin cytotoxicity and demonstrated a robust inducible response to cisplatin treatment.

### p38 MAPK downstream effectors are needed for survival on cisplatin

We hypothesized that mutants in some of the p38 MAPK-dependent immunity genes identified in our proteomics analysis would have an impact on cisplatin sensitivity. We tested the cisplatin sensitivity of mutants in five of these genes: *catp-3, clec-4, clec-66, dod-24,* and *lys-1*, for which mutants were available. All five are categorized as immune response genes. Remarkably, single deletion mutants in four of these genes (*catp-3, clec-4, dod-24,* and *lys-1*) showed greatly decreased survival upon cisplatin exposure (Fig. 8c). Moreover, the qRT-PCR analysis for *clec-4, catp-3, dod-24,* and *lys-1* confirmed that there was cisplatin-dependent induction of these immune effectors at the transcriptional level as well (Fig. 8d). Transcript accumulation of all four genes were also induced in the absence of the bacteria (Supplementary Fig. 14a) proving further that the response to cisplatin is not an effect produced by bacteria. Next, we found that induction of *clec-4, catp-3, dod-24,* and *lys-1* was not observed upon cisplatin exposure in the absence of PMK-1 (Supplementary Fig. 14b). This further confirmed the importance of these genes for cisplatin resistance and their dependence on the p38 MAPK signaling cascade. Induction of *clec-4, catp-3, dod-24,* and *lys-1* was also not observed in *ire-1* mutants following cisplatin exposure, showing that they operated downstream and not in parallel to IRE-1 (Supplementary Fig. 14c). Taken together our work shows that IRE-1/p38 MAPK/ATF-7 mediated gene expression response in intestinal cells confers protection to *C. elegans* somatic tissues against cisplatin cytotoxicity (Fig. 9). Surprisingly, activation of typeII detoxification genes via SKN-1 seemed not to play an important role in protection from cisplatin cytotoxicity in post-mitotic adults (Fig. 9).

### Cisplatin exposure leads to the accumulation of necrotic damage

Multiple mechanisms are thought to be the cause of cisplatin-induced death. To investigate the nature of cisplatin-induced cell death we used apoptosis, necrosis, and autophagy markers to assess the cause of death of cisplatin-treated worms. We did not observe the increase in

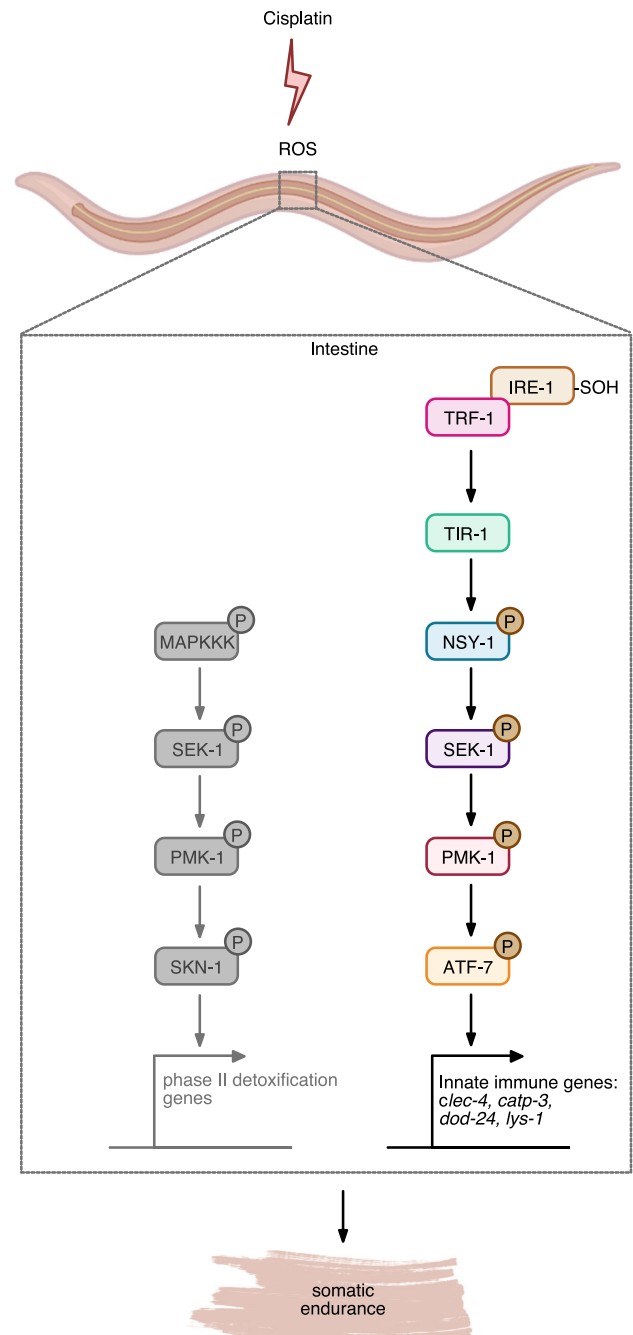

**Fig. 9 | Innate immune response in intestinal cells confers protection of *C. elegans* somatic tissues against cisplatin cytotoxicity.** Working model for p38 innate immune pathway modulation of cisplatin response in adult *C. elegans*. Based on our data, we propose that the p38 immune signaling pathway was activated in the intestine via IRE-1/TRF-1 to drive transcription of ATF-7-dependent immune genes and promote somatic endurance (right pathway). Activation of detoxification genes via SKN-1 transcription factor was not sufficient to promote survival (left pathway, marked in gray). Created with BioRender.com.

apoptotic corpses marked by CED-1::GFP in somatic tissues in cisplatin-treated *clec-4(ok2050)* mutants in comparison to untreated control (Supplementary Fig. 15a). No increase was detected in the number of puncta of intestinal mCherry::LGG-1[9] in *clec-4(ok2050)* mutants upon cisplatin treatment indicating the absence of autophagy induction (Supplementary Fig. 15b). We chose intestine-expressed mCherry::LGG-1 for our analysis since the intestine was important for cisplatin response. We did however observe an increase in vacuole number in

*clec-4 (ok2050)*, *catp-3(ok1612)*, and *dod-24(ok2629)* mutants upon cisplatin treatment at levels higher than that seen in wild-type controls (Supplementary Fig. 16a, b). Such vacuoles are characteristic of necrotic death[38]. Importantly in all cases, necrotic vacuoles were found in intestinal cells (Supplementary Fig. 16a). Necrotic vacuoles are characterized by the presence of phosphatidylserine (PS) on the outer leaflet of the vacuolar membranes. The surface exposure of phosphatidylserine can be tracked by the binding of a soluble MFG-E8:mCherry[39] protein marker (Supplementary Fig. 16c), which binds PS in the vacuolar membranes of necrotic cells[40]. We found that the vacuoles induced at high levels in *dod-24(ok2629)* and *catp-3(ok1612)* mutants had PS-positive surfaces, thus providing further evidence for identification of the vacuoles as necrotic vacuoles. Neurotoxic and nephrotoxic side effects of cisplatin remain the major limitations in the use of the drug. It has been previously shown that low doses of cisplatin did not affect the basal and touch-provoked movement of the nematode and did not cause broad damage to sensory functions of *C. elegans*[41]. Therefore, we excluded the neurotoxicity as a likely cause of death of the mutants. We focused on the possibility of excretory system dysfunction as a cause of cisplatin-induced death. We tested the cisplatin sensitivity of the mutants with excretory canal defects: *exc-5(rh232)*, *exc-6(rh103)*, and *exc-7(rh252)*. All three mutants displayed significantly decreased cisplatin survival (Supplementary Fig. 17a). EXC-4p::EXC-4::GFP marks the membranes of the excretory tubules. No differences were however observed in the GFP localization pattern between cisplatin-treated and untreated animals (Supplementary Fig. 17b). We concluded that the defect in excretory canal function is lethal for cisplatin-treated worms but the exposure of adults to cisplatin will not cause degeneration of the excretory canals.

## Discussion

We have described the identification and characterization of the p38 MAPK immune pathway as a necessary component in cisplatin detoxification and protection from cisplatin toxicity in post-mitotic cells of *C. elegans*. Here we show that the induction of type II detoxification genes by SKN-1/NRF upon cisplatin exposure provoked by an increase in ROS levels of post-mitotic cells was not important for cisplatin resistance. Rather, we show that PMK-1-mediated innate immunity response via the ATF-7/ATF2 transcription factor was a major component in the resistance against cisplatin. The finding that *tir-1(qd4)* mutants, which only have a documented role in innate immunity, are also cisplatin sensitive provides further evidence for a role for innate immunity pathway in mediating resilience. Furthermore, cisplatin exposure led to the phosphorylation of PMK-1 and ATF-7 and establishes an early molecular landmark for cisplatin exposure. We found that the IRE-1/TRF-1 module acts upstream of the NSY-1/SEK-1/PMK-1 pathway to activate signaling. Cisplatin induced ROS was needed for this signaling cascade activation and cisplatin provoked sulfenylation of IRE-1 was observed. Finally, we identified four p38 MAPK-dependent genes whose function is necessary for resilience thus completing a link from cisplatin exposure, through activation of a signaling cascade to the function of individual effectors at the end of the cascade. We found increased levels of necrotic damage in the identified immune mutants thus providing a possible cellular basis for the cisplatin cytotoxicity.

Cisplatin is one of the most effective anticancer drugs and its effectiveness seems to be due to the unique properties of cisplatin and its ability to enter cells via multiple pathways. One possible killing mechanism of cisplatin in dividing cells is through the creation of DNA crosslinks, which results in the arrest of cell division and growth and triggers apoptosis[1]. In nuclei, cisplatin forms adducts with two consecutive guanines or adenines. This modified bond prevents DNA repair, which results in cell death[1]. *C. elegans* has been widely used to study the response of dividing cells to cisplatin. This includes cisplatin-induced DNA-damage response pathways involved in DNA inter-strand

crosslinks repair[42] and pathways ensuring the proper replication of damaged DNA in embryos[43]. Analysis of the role of cisplatin in DNA damage is important for its role in anti-tumor therapy. However, while interaction with DNA in nuclei results in cell cycle arrest and initiates cell death in proliferating cells, only 1–10% of intracellular cisplatin is found in nuclei[3,4,44].

A major drawback in anti-cancer treatment against solid tumors is the presence of dormant malignant cells responsible for tumor relapse. Such non-dividing cells are resistant to standard chemotherapy agents that rely on DNA replication and cell division to promote their antitumor effect. The changed phenotype of quiescent cells enables them to survive the chemotherapeutic treatment[45]. In our study, we have used cisplatin-resistant adult *C. elegans* worms as a model organism to study the resistance mechanism of non-dividing cells. The fates of every cell in the nematode have been determined and all the somatic cells of an adult worm are terminally differentiated and non-dividing[46,47]. Therefore, *C. elegans* is a good model for the analysis of cisplatin resistance in non-dividing cells. Changes in gene expression have been quantified between cisplatin-resistant and sensitive cell tumor lines and these do not always correlate well with the protein abundance[48–50]. With that in mind, we performed a quantitative proteomics screening of cisplatin-resistant wild-type *C. elegans* to identify the proteins that display differential abundance in response to cisplatin and to characterize factors responsible for the resistance.

Our previous work has demonstrated in worms that there is an increase in ROS levels when *C. elegans* 1-day-old adults are subjected to cisplatin treatment. This consequently led to higher levels of oxidized ASNA-1 and disturbances in ER-targeted tail-anchored membrane protein insertion[9,51]. In this study, we started by asking to what extent ROS detoxification was essential for survival. For this, we tested the impact of the stress-activated p38 MAPK pathway on the resistance of post-mitotic worms to cisplatin treatment. MAPKs are critical components of intracellular networks which regulate gene expression in response to a range of extracellular stimuli including ROS. p38 MAPK pathway activation is an important factor in the cellular response to cisplatin[52]. In *C. elegans* activation of the SKN-1 transcription factor by the p38 MAPK pathway in the intestine is the first line of defense against xenobiotics[21] via regulation of oxidative stress resistance[23,53]. SKN-1 is also involved in innate immunity response to pathogen infection[54]. Knockdown or inhibition of mammalian homolog of SKN-1, NRF2, causes increased cell death, cytotoxicity, and apoptosis in response to cisplatin treatment[55,56]. SKN-1 and DAF-16 are key components in the response of *C. elegans* developing larvae containing proliferative cells to cisplatin[27]. In this study, we examined whether SKN-1 also regulates the response to cisplatin in non-dividing cells. Many phase II detoxification genes which are regulated by SKN-1 are also upregulated upon cisplatin exposure. Surprisingly, *skn-1* mutants did not exhibit any sensitivity phenotype to cisplatin treatment, although their phenotype in sensitivity factor to other pro-oxidants like arsenite[21] or paraquat[23] is well established. Furthermore, DAF-16/FOXO which is an oxidative stress-induced factor with a well-established involvement in protection from oxidative stress generators like arsenite[57] or paraquat[26], also had no role in protection from cisplatin.

Hourihan et al.[36] have shown that endoplasmic reticulum stress sensor IRE-1 has a UPR^ER independent redox-regulated function[36]. Localized ROS activation leads to IRE-1 sulfenylation, which in turn inhibits IRE-1-mediated UPR^ER. IRE-1 works with TRF-1 to initiate the antioxidant response via activation of the p38 MAPK pathway through SKN-1/NRF activity. Using these findings as a starting point, we found that IRE-1 and TRF-1 are also important upstream activators of the p38 MAPK pathway in the context of cisplatin exposure. Moreover, that response was independent of the UPR^ER activity of IRE-1. In contrast, SKN-1/NRF driven antioxidant response was not important in protection from cisplatin toxicity. Rather our finding showed that the IRE-1/

TRF-1/p38 MAPK cascade activated ATF-7-dependent immune response genes.

Sterile inflammation refers to inflammatory reaction pathogens other than pathogens. This process is activated by damage-associated molecular patterns (DAMPs) produced by cell or tissue damage[58]. DAMPs can initiate the immune response through the activation of pattern recognition receptors (PRPs) which include Toll-like, NOD-like, and C-type lectin receptors[59]. Though sterile inflammation plays a crucial role in tissue repair, unresolved chronic inflammation may lead to inflammatory or neurogenerative diseases as well as autoimmune diseases and cancer. Survival following traumatic brain injury in *Drosophila* was increased by induction of NF-κB innate immune response transcription factor Relish[19]. Moreover, macrophage-inducible C-type lectin (MINCLE), was shown to induce pathogenic proinflammatory responses in experimental traumatic brain injury and ischaemic stroke[60,61]. One explanation for our findings is that cisplatin exposure of post-mitotic *C. elegans* will lead to the induction of DAMPs and therefore initiate innate immune response through the activation of PRPs. We indeed observed in our proteomics analysis an increased immune response including several C-type lectins such as CLEC-4 and CLEC-66. The expression of many of those host defense proteins was dependent on SEK-1 confirming findings from previous studies in bacterially infected worms[12]. Interestingly, it has been shown that there is upregulation of the key components of immune response to infection, NFκB or TNF, in cisplatin-resistant lung cancer cell lines. Moreover, the inhibition of those two components increase cisplatin sensitivity in resistant cell lines[62]. We found that the deletion of single p38-dependent response genes resulted in a strong cisplatin sensitivity phenotype. Our data suggest a mechanism by which the knockdown of immune genes regulated by the p38 MAPK cascade in resistant post-mitotic cells leads to increased cisplatin sensitivity. A possible mechanistic explanation lies in the fact that three of the identified proteins LYS-1, DOD-24, and CLEC-4 are predicted with high confidence to be secreted proteins[63]. This suggests that such defense proteins secreted from the intestine might spread systemically to protect nearby tissues. LYS-1 is also activated by bacterial infection and by exposure to xenobiotic graphene oxide. In both cases, LYS-1 acts upstream of TUB-1/TULP4 signaling[64,65], which is involved in the proteasomal degradation of target proteins in response to stress. The DOD-24 protein contains a CUB domain which is present in secreted proteins associated with tissue repair and inflammation. Knockdown of the murine homolog of CATP-3 (ATP4a) leads to secretory defects in intestinal cells and reduced levels of secretory granules[66]. We hypothesize that even though CATP-3 is a predicted integral membrane protein, its loss could lead to secretion defects as well. Treatments that target the immunity pathways in resistant tumors should be explored as possible means to combat cisplatin resistance in non-dividing cells.

## Methods

Ethical permissions were not needed for this work since it was performed using an invertebrate model system. No local rules are applicable to research on invertebrates. Husbandry and procedures were performed as appropriate for the species.

### *C. elegans* strains maintenance and synchronization

PHX2311: *sek-1(syb2311)* was obtained by precise deletion of the entire coding sequence of *sek-1* using CRISPR-CAS9 technology by Suny Biotech. PHX2456: *sek-1(syb2456)* was generated using CRISPR-CAS9 technology by precise insertion of the flexible linker GASGASAS (GGAGCATCGGGAGCCTCAGGAGCATCG), cMyc, and AID just before the stop codon. PHX4220: *sek-1(syb4220)* was generated using CRISPR-CAS9 technology by precise insertion of the GASGASAS flexible linker, 3xFlag, and AID just before the stop codon of the gene. PHX5891: *ire-1(syb5891)* was generated using CRISPR-CAS9 technology to precisely

insert the GASGASGAS flexible linker, and 3xFlag just before the stop codon. PHX6493: *ire-1(syb6493)* was generated by using CRISPR-CAS9 technology by precise mutation of cysteine (TGT) to serine (TCG) at amino acid 663 at the IRE-1 locus in *ire-1(syb5891)*. All *C. elegans* strains used in this study are listed in Supplementary Data 5. Worms were cultured under standard conditions at 20 °C on nematode growth media (NGM) plates[67] unless stated otherwise and the *E. coli* strain OP50 was used as a food source. OP50-GFP was obtained from the *Caenorhabditis* stock center. All worms used in the study were synchronized to the young adult stage. Synchronous larval populations were obtained by gravity separation as described[68]. The L1 larvae were cultured for three more days at 20 °C to obtain populations with synchronous worms at the 1-day-old young adult stage.

### Western blot analysis

Western blots were performed as described previously[69]. Primary antibodies were: anti:phospho-p38 MAPK (Thr180/Thr182) Rabbit mAb (D3F9, Cell Signaling) 1:1000, anti:total PMK-1 MAPK Rabbit pAb 1:1000 from the Pukkila-Worley lab[70], anti:tubulin (T5168, Sigma) 1:5000, anti:Flag (M2, Sigma) 1:1000.

### Western blot analysis with SuperSep Phos-tag gels

SuperSep Phos-tag, 12.5%, 17 well gels were from Fuji Film Wako Pure Chemical Corporation (#195-17991). Sample preparation and electrophoresis were carried out using the same procedure as for regular western blot experiments except that before transfer to PVDF membranes, the gels were gently agitated in a transfer buffer containing 10 mmol/L EDTA for 1 h as per the manufacturer's protocol. It is important to note that Phos-tag gels cause molecular weight ladders to migrate differently and the migration pattern is not indicative of the actual protein size.

### Sulfenylation assay

Assay was performed as described previously[36]. In short, after experimental treatment, nematodes were lysed in lysis buffer (50 mM HEPES, 50 mM NaCl, 1 mM EDTA, 10% glycerol, 1% Triton x-100) supplemented with 1 mM DCP-Bio1, 0.1 mM N-ethyl malemide, 0.1 mM Iodacetamide, and protease inhibitor. Homogenization was done using the Next Advance cell disruptor and 0.2 mm stainless steel beads for 3 min at 4 °C. Next, samples were centrifuged at $14,000 \times g$ for 20 min at 4 °C. The supernatant was transferred to a new tube and incubated in the dark for 1 h at room temperature to allow for the labeling of sulfenic acids. After incubation, un-reacted DCP-Bio1 was removed using P6-Spin Columns (#7326227, BioRad) following the manufacturer's instructions. Protein concentrations were determined by the BCA assay and 600–800 µg of total protein was added to a 50 µl slurry of Pierce Streptavidin Magnetic Beads (#88816, Thermo Fisher Scientific) in a total volume of 900 µL. The mix was rotated for 1 h at room temperature. After the binding, the beads were magnetically separated from the lysate, the supernatant was discarded and the beads were washed three times for 10 min each in 1 ml of binding/wash buffer. Next, beads were separated and heated for 10 min at 95 °C in 30 µl of reducing 2 × LDS buffer (#1610737, BioRad). The beads were magnetically separated and the total lysate was analyzed by electrophoresis using SDS-PAGE followed by western blotting.

### Detection of carbonylated protein

Protein carbonylation was determined by OxyBlot Protein Oxidation Detection Kit (S7150, Merck Millipore). Homogenization and protein estimation were done as described previously[69]. Total extracted proteins were derivatized according to the manufacturer's instructions. 2-mercaptoethanol was used to reduce samples. The membrane was blocked for 1 h in 5% BSA/TBS-T and probed with antibodies provided in the kit according to the manufacturer's instructions.

## RNA isolation and quantitative RT-PCR

RNA isolation was performed as described previously[9]. qPCR was performed on a CFX Connect (BioRad) instrument using KAPA SYBR FAST qPCR Kit (KK4600, Sigma) with normalization to the housekeeping gene F44B9.5. Analysis was performed using the comparative Ct method All samples were tested in triplicates. Data on the graph is presented on a logarithmic scale. Oligonucleotides sequences are provided as a Supplementary Table 1.

## Fluorescence microscopy and pixel intensity measurement

Live 1-day-old adult animals were sedated in 2 mM Levamisole/M9 and mounted onto 2% agarose pads. ImageJ was used to quantify pixel intensity in worms.

## Confocal microscopy

Live 1-day-old adult animals were sedated in 2 mM Levamisole/M9 and mounted onto 2% agarose pads. The fluorescence signals were analyzed at 555 nm using a LSM700 Confocal Laser Scanning Microscope (Carl Zeiss) with LD C-Apochromat 63x/1.4 Oil DIC objective.

## Cisplatin treatment

Worms were treated with cisplatin (i) on plates or (ii) in liquid. (i) Cisplatin plates were prepared using MYOB media with 2% agar in which the drug was added at a final concentration of 300 or 600 µg/mL. Cisplatin solution (1 mg/mL, Accord Healthcare AB) was added to the autoclaved medium after cooling to 54 °C. 1-day-old adults were collected, washed, and incubated on cisplatin plates for 3 or 6 h. Concentrated OP50 was used as a food source. (ii) 1-day-old adults were washed extensively in M9 before analysis in order to remove the traces of the live bacteria from the surface of the worms as well as from the intestine of the worms. Washed worms were exposed to cisplatin solution (300 µg/mL for 2 h or 3.5 h) by rotation at room temperature thus allowing for uniform exposure to the drug.

## Auxin treatment

1-day-old adult animals were transferred to NGM plates containing 1 mM auxin (indole-3-acetic acid) (#A10556.06, Alfa Aesar) for indicated times. The 1 mM auxin-containing agar plates were prepared 24 h before use from a freshly made 800 mM stock solution in 99.5% ethanol and stored in the dark. An equivalent amount of ethanol was added to the control plates.

## Pro-oxidant treatment

Worms were staged by gravity and L1 larvae were collected. L1 larvae were grown on NGM plates with or without 0.5 mM MitoTempo (#SML0737, Sigma) for 72 h until they reached adulthood. 1-day-old adult worms were washed extensively with M9 to remove any traces of the bacteria and exposed with rotation for 2 h to either M9 containing 0.5 mM MitoTempo or to cisplatin solution (300 µg/mL) diluted from the stock in M9 and containing 0.5 mM MitoTempo.

## Cisplatin sensitivity assay

Cisplatin sensitivity assay was performed as described previously[9]. For worms pretreated with 1 mM auxin and subjected to cisplatin sensitivity assay, the cisplatin-containing agar plates also contained 1 mM auxin.

## Proteomics sample preparation

1-day-old synchronized animals without or with cisplatin (300 µg/mL) treatment on plates for 6 h were lysed in 50 mM TEAB and 2% SDS and protein determination was performed. Three representative references pools were made by taking equal protein amounts from wild-type and mutant samples and combine into a wild-type, a mutant, and a wild-type/mutant reference. The samples were processed according to the filter-aided sample preparation (FASP) method modified from Wisniewski et al.[71]. Briefly, the samples and references (30 µg) were reduced with 100 mM dithiothreitol at 56 °C for 30 min, transferred to 30 kDa centrifugal filters (Pall Nanosep, Sigma), washed several times with 8 M urea, and once with digestion buffer (DB, 0.5% sodium deoxycholate in 50 mM triethylammonium bicarbonate) prior to alkylation with 10 mM methyl methanethiosulfonate in DB for 20 min at room temperature. Digestions were performed by the addition of 0.3 µg Pierce MS grade Trypsin (Thermo Fisher Scientific) in DB and incubated overnight at 37 °C. An additional portion of trypsin (0.3 µg) was added and incubated for another three hours. Peptides were collected by centrifugation and labeled using TMT 10-plex isobaric mass tagging reagents (Thermo Fisher Scientific) according to the manufacturer´s instructions. The wild-type and mutant samples were analyzed in separate sets and included two references each. The labeled samples were combined and sodium deoxycholate was removed by acidification with 10% trifluoroacetic acid. The combined sample was desalted using Pierce Peptide Desalting Spin Columns (Thermo Fisher Scientific) according to the manufacturer's instructions. The proteins were pre-fractionated into 40 fractions by basic reversed-phase chromatography (bRP-LC) using a Dionex Ultimate 3000 UPLC system (Thermo Fisher Scientific). Peptide separation was performed using a reversed-phase XBridge BEH C18 column (3.5 µm, 3.0 × 150 mm, Waters Corporation) and a linear gradient from 3 to 40% acetonitrile in 10 mM ammonium formate buffer at pH 10.00 over 17 min, followed by an increase to 90% acetonitrile over 5 min at a flow of 400 µl/min. The fractions were concatenated into 20 fractions, dried, and reconstituted in 3% acetonitrile, and 0.2% formic acid.

## nanoLC-MS analysis and database matching

The fractions were analyzed on an Orbitrap Fusion Tribrid mass spectrometer interfaced with Easy-nLC1200 liquid chromatography system (both Thermo Fisher Scientific). Peptides were trapped on an Acclaim Pepmap 100 C18 trap column (100 µm × 2 cm, particle size 5 µm, Thermo Fisher Scientific) and separated on an in-house packed analytical column using a gradient from 5 to 80% acetonitrile in 0.2% formic acid over 80 min followed by 80% acetonitrile for 10 min at a flow of 300 nL/min. MS scans were performed at a resolution of 120,000, $m/z$ range 380–1380. MS2 analysis was performed data-dependent, with a top speed cycle of 3 s for the most intense e precursor ions with a charge state of 2–7. Precursor ions were isolated in the quadrupole with a 0.7 $m/z$ isolation window and dynamic exclusion of fragmented precursors was set to 10 ppm for a duration of 60 s. Isolated precursor ions were subjected to collision-induced dissociation (CID) with collision energy set to 35 and a maximum injection time of 50 ms. The MS2 fragment ions were detected in the ion trap followed by multi-notch (simultaneous) isolation of the top 5 abundant fragment ions for further fragmentation (MS3) by higher-energy collision dissociation (HCD) at 65% and detection in the Orbitrap at 50,000 resolutions. Protein identification and quantification were performed using Proteome Discoverer version 2.2 (Thermo Fisher Scientific). The search was performed by matching against *Caenorhabditis elegans using* SwissProt (November 2017) using Mascot 2.5.1 (Matrix Science) with a precursor mass tolerance of 5 ppm and fragment mass tolerance of 0.6 Da. Tryptic peptides were accepted with zero missed cleavage, variable modifications of methionine oxidation, and fixed cysteine alkylation; TMT-label modifications of N-terminal and lysine were selected. Percolator was used for PSM validation with a strict FDR threshold of 0.01. TMT reporter ions were identified in the MS3 HCD spectra with 3 mmu mass tolerance, and the TMT reporter intensity values for each sample were normalized on the total peptide amount. The references were used as the denominator. Only unique identified peptides were considered for the relative quantification and quantified proteins were filtered at 1% FDR.

## Statistical analysis of proteomics data

The differential expression analysis was performed in R. Differentially expressed proteins were identified by using a two-sided *t*-test on

log-transformed data. To control for multiple testing, the Benjamini-Hochberg procedure was used. Proteins with an FDR value < 0.05 were considered differentially expressed. Principal component analysis (PCA) was used as quality control for the samples and clustering of groups. Enrichment analysis of the differentially expressed proteins was subsequently performed using the Reactome database.

## Statistical analysis

Statistical analysis was performed with Prism 7 software (Graph-Pad software, La Jolla,). $P$-values indicated statistical significance (*$p < 0.05$, **$p < 0.01$, ***$p < 0.001$).

## Reporting summary

Further information on research design is available in the Nature Portfolio Reporting Summary linked to this article.

## Data availability

All data is available in the main text or supplementary materials. Source data are provided with this paper. The mass spectrometry proteomics data have been deposited to the ProteomeXchange Consortium via the PRIDE[72] partner repository with the dataset identifier PXD033377. All other data are available in the article and its Supplementary files or from the corresponding author upon request. Source data are provided with this paper.

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

## Acknowledgements

We thank the Caenorhabditis Genetic Center (funded by NIH Office of Research Infrastructure Programs P40 OD010440) and National Bioresource Project for the Experimental Animal "Nematode C. elegans" for providing strains, Proteomics Core Facility of Sahlgrenska Academy, University of Gothenburg for proteomic analysis, Read Pukkila-Worley, Dave Reiner & Ji Ying Sze for providing strains and Read Pukkila-Worley for providing anti:total PMK-1 antibody, and Pinar Barkan for discussions on proteomics approaches. The work was supported by grants from the Swedish Cancer Society CAN 2021/1807 Pj (P.N.) and ALF means nr: ALFGBG-966362 (P.N.); Stiftelsen Assar Gabrielssons Fond FB19-44 (D.R.) and Stiftelsen Assar Gabrielssons Fond FB20-32 (D.R.).

## Author contributions

D.R. and G.K. designed experiments. D.R., B.K., and G.K. performed the experiments. D.R., A.P.-F., J.M., and G.K. analyzed the data. D.R. wrote the paper. G.K. and P.N. critically revised the manuscript for intellectual content. All authors approved the final version.

## Funding

## Competing interests

The authors declare no competing interests.
