## [Peer Review File · Nature Communications]

Cisplatin toxicity is counteracted by the activation of the p38/ATF-7 signaling pathway in post-mitotic *C. elegans*REVIEWER COMMENTS

Reviewer #1 (Remarks to the Author):

Though an effective chemotherapeutic drug targeting dividing cells, cisplatin efficacy is reduced in non-mitotic tissue. Furthermore, its use is sometimes limited by side effects caused by damage to post-mitotic cells such as those in the nervous system or kidney. The mechanism of action in post-mitotic cells has been heretofore unknown. Raj and colleagues here fill this gap in our knowledge by identifying a signal transduction cascade as well as down-stream effectors that play a role in cisplatin induced toxicity. The authors make powerful use of the *C. elegans* experimental system. Using a simple survival assay (but see below concern about cause of death), they assess roles of various genes in cisplatin toxicity. They nicely combined the genetics with omic approaches.

They implicate the innate immune response activation and suggest that “Sterile inflammation” plays a role in cisplatin toxicity. They identify down-stream effectors using a quantitative proteomic approach.

I have minor concerns mostly related to the writing (see below). A more substantive concern relates to their conclusion of sterile inflammation. The alternative hypothesis, which they acknowledge and attempt to address, is that cisplatin injures epithelia but it is ultimately the bacteria that cause the demise of the animal and the activation of the immune response. They argue against this by noting an absence of bacterial proliferation in intestinal cells. But a simple experiment they should do is ask whether cisplatin still triggers innate immune response genes in animals that are fed dead bacteria. I also would like to know how the animals die. Is it due to excretory dysfunction, which would parallel human nephrotoxicity? Or is it due to neuron toxicity, which would parallel human peripheral neuropathy? This is important in considering the mechanism by which activation of the innate immune response protective against cisplatin. For example, does the disabling of the immune response lead to excessive protein oxidation shown in fig S1? Is there literature (or can the authors speculate based on protein domain structures) that could explain how *clec-4*, *catp-3*, *dod-24*, and *lys-1* protect against CP-induced death? I would NOT expect the authors to elucidate the mechanism by which immune target genes protect against CP but would like to see some discussion of this point.

Other (more minor) specific points

1. The writing could be improved. Pay close attention to tense and to writing in the active voice. Passive voice is much harder to read. An example from the abstract “Innate immune response proteins whose increased abundance depended on SEK-1 activity and cisplatin exposure were identified.” Write instead “We identify Innate immune response proteins whose increased abundance depended on SEK-1 activity and cisplatin exposure”.
2. The following sentence in the abstract needs a verb “We conclude that in contrast to larvae with proliferating cells, in adult post-mitotic tissues, protection from ROS-induced damage not important.”
3. Avoid injudicious use of adverbs such as “extremely sensitive” and “very important”. Simple say “sensitive” and “important”
4. It would be useful to know how an LD50 of 150 micrograms/mL relates to the LD50 of wild type animals. Fig S2 only goes up to 200.
5. Can you use AID system to assess where *sek-1* is required? compare an intestinal to hypoderm TIR1.

Reviewer #2 (Remarks to the Author):

Cisplatin is a widely used chemotherapeutic drug for cancer treatment. However, it is known to kill both proliferating cells and post-mitotic cells (such as inner ear hair cells and kidney cells), which rules out its applications in some vulnerable patients due to the damages to these organs. Thus, it is very important to understand the underlying mechanisms by which cisplatin kills non-dividing cells. In this study, Raj et al. used *C. elegans* as a model to investigate the mechanisms of cisplatin

chemoresistance because all somatic cells in adult *C. elegans* are post-mitotic. Combining *C. elegans* genetics with proteomic analyses and stress assays, the authors showed that the innate immune response branch downstream of PMK-1/P38 MAPK is critical for the cisplatin chemoresistance. Specifically, they revealed that the transcription factor ATF-7 and multiple downstream immune effectors play essential role in the killing effect of cisplatin on post-mitotic cells. Overall, this study was well conducted, and the findings are important. Below are my remaining critiques:

- 1) 24 hour cisplatin treatment is sufficient to kill many innate immunity mutant worms. How are these worms killed in general? Some imaging experiments could be done to elaborate the nature of cisplatin-induced cell death. Apoptosis, necrosis, or other forms of cell death? Does cell death only occur to the gut or other tissues as well?
- 2) For inner ear hair cells, there are studies showing that ROS and detoxification systems are essential for the cisplatin toxicity. But this does not seem to be the case in this study, as SKN-1 and detox system are clearly not required. How to reconcile this discrepancy?
- 3) In Fig 5a, the survival result of *pmk-1(km25)* mutant and *atf-7(qd22)* mutant seems additive. If *atf-7* mediates *pmk-1*'s role in cisplatin chemoresistance, should double mutant behave similarly to the *atf-7* single mutant? An additive effect would suggest two parallel pathways to me.
- 4) The P38 MAPK pathway is very conserved across species. Does the current finding apply to mammals? The authors can at least verify their results using mammalian cell culture systems.

Reviewer #3 (Remarks to the Author):

Raj and colleagues present a manuscript describing resistance of post-mitotic cells to cisplatin exposure that is dependent on the p38 pathway in *C. elegans*. Using characterized loss-of-function mutant alleles in p38 pathway components, the authors demonstrate that the signaling cascade of TIR-1/NSY-1/SEK-1/PMK-1 is required for resistance against cisplatin toxicity. This response they show is independent of the downstream transcription factor SKN-1 and instead dependent on the transcription factor ATF-7. By depleting SEK-1 using a somatic tissue specific degron, the authors demonstrate that SEK-1 is required in somatic adult cells for cisplatin resistance. Unbiased proteomics of cisplatin exposed animals revealed many differentially regulated proteins, including innate immune effectors. The authors demonstrate that cisplatin exposure transcriptionally induces three innate immune effector genes in a p38 dependent manner. Additionally, the authors provide data that suggests that cisplatin exposure induces phosphorylation of SEK-1 and ATF-7. Using proteomics, the authors further identified 22 proteins that are induced by cisplatin in a *sek-1* dependent manner. Finally, they demonstrate that loss-of-function mutants in three proteins identified in their proteomics analysis are required for cisplatin resistance.

The data presented by the authors that the p38 PMK-1 pathway is required for protection against cisplatin toxicity is convincing. However, I do not think that this manuscript presents enough of a mechanistic advance to be appreciated by a general audience. It is well described in the *C. elegans* literature that the p38 signaling pathway is required for resistance against a diverse number of stressors (e.g., toxins, infection, heat, and osmotic). Thus, it is not surprising that the p38 pathway is also required for cisplatin resistance. Furthermore, it has been previously documented in the mammalian literature that the p38 pathway is required for cisplatin resistance. It is also important to note that the authors did not demonstrate that cisplatin induces p38 phosphorylation, and instead rely on transcriptional studies of three of seven genes. Also of note, many of the most significantly regulated proteins in their proteomics experiment are components of the cytosolic and endoplasmic reticulum unfolded protein responses, rather than innate immune effectors.

In my view, two key questions are not addressed in this manuscript, which would considerably elevate the impact of the story. First, how is the p38 pathway activated during cisplatin exposure? Second, how are the genes mobilized by the p38 pathway during cisplatin exposure providing protection against this toxic agent? In the absence of these insights, I do not think that this manuscript rises to a level appropriate for publication in *Nature Communications*.

Finally, I also have concerns about the central claim of the paper that the innate immune system promotes cisplatin chemoresistance. The authors make this claim based on the fact that individual genes dependent on the p38 pathway in the response to cisplatin have been annotated as “innate immune” genes. In of itself, however, this does not provide a functional connection between innate immunity and cisplatin resistance, as these effectors could have multiple functions in different contexts.

Here we provide the detailed point by point response to concerns raised by the reviewers:

REVIEWER #1

*Though an effective chemotherapeutic drug targeting dividing cells, cisplatin efficacy is reduced in non-mitotic tissue. Furthermore, its use is sometimes limited by side effects caused by damage to post-mitotic cells such as those in the nervous system or kidney. The mechanism of action in post-mitotic cells has been heretofore unknown. Raj and colleagues here fill this gap in our knowledge by identifying a signal transduction cascade as well as down-stream effectors that play a role in cisplatin induced toxicity. The authors make powerful use of the *C. elegans* experimental system. Using a simple survival assay (but see below concern about cause of death), they assess roles of various genes in cisplatin toxicity. They nicely combined the genetics with omic approaches.*

They implicate the innate immune response activation and suggest that “Sterile inflammation” plays a role in cisplatin toxicity. They identify down-stream effectors using a quantitative proteomic approach.

I have minor concerns mostly related to the writing (see below). A more substantive concern relates to their conclusion of sterile inflammation. The alternative hypothesis, which they acknowledge and attempt to address, is that cisplatin injures epithelia but it is ultimately the bacteria that cause the demise of the animal and the activation of the immune response. They argue against this by noting an absence of bacterial proliferation in intestinal cells. But a simple experiment they should do is ask whether cisplatin still triggers innate immune response genes in animals that are fed dead bacteria.

*I also would like to know how the animals die. Is it due to excretory dysfunction, which would parallel human nephrotoxicity? Or is it due to neuron toxicity, which would parallel human peripheral neuropathy? This is important in considering the mechanism by which activation of the innate immune response protective against cisplatin. For example, does the disabling of the immune response lead to excessive protein oxidation shown in fig S1? Is there literature (or can the authors speculate based on protein domain structures) that could explain how *clec-4*, *catp-3*, *dod-24*, and *lys-1* protect against CP-induced death? I would NOT expect the authors to elucidate the mechanism by which immune target genes protect against CP but would like to see some discussion of this point.*

Other (more minor) specific points

- 1. The writing could be improved. Pay close attention to tense and to writing in the active voice. Passive voice is much harder to read. An example from the abstract “Innate immune response proteins whose increased abundance depended on SEK-1 activity and cisplatin exposure were identified.” Write instead “We identify Innate immune response proteins whose increased abundance depended on SEK-1 activity and cisplatin exposure”.*
- 2. The following sentence in the abstract needs a verb “We conclude that in contrast to larvae with proliferating cells, in adult post-mitotic tissues, protection from ROS-induced damage not important.”*
- 3. Avoid injudicious use of adverbs such as “extremely sensitive” and “very important”. Simple say “sensitive” and “important”*
- 4. It would be useful to know how an LD50 of 150 micrograms/mL relates to the LD50 of wild type animals. Fig S2 only goes up to 200.*
- 5. Can you use AID system to assess where *sek-1* is required? compare an intestinal to hypoderm TIR1.*

REVIEWER #1 RESPONSE:

We would like to thank reviewer #1 for raising concerns about innate immunity response being caused by feeding bacteria rather than the cisplatin exposure. We indeed tried to exclude the concern about the bacteria being the cause of the innate immune response with experiment where GFP:OP50 was used. Experiments with GFP labeled bacteria have been used to address the concern of bacterial infection in *C. elegans* intestine (PMID: 30965033). Here we failed to detect any GFP labeled bacteria in the intestinal cells of drug-exposed worms or any damage to the intestinal cells in the time period when immune response genes are induced in the intestine (**Fig. 4b**). Moreover, we confirmed that the *E. coli* was killed

within 6 hours of cisplatin exposure. We thought that those experiments provided us with sufficient evidence for lack of a bacterial effect on the innate immune genes activation. The reviewer also suggested we should ask whether cisplatin still triggers innate immune response genes in animals that are fed dead bacteria. Unfortunately, this experimental approach is not going to answer the question asked by the reviewer. This is because heat-killed OP50 bacteria lacks certain nutrients or molecules that are required for larval growth (PMID: 28569665). Moreover, nutrient deficiency in heat-killed-OP50 induces a protective response from the worms similar to starvation response (PMID: 28569665). This observation was consistent with another study suggesting that *E. coli* contains heat-labile nutrients required for *C. elegans* normal growth and longevity (PMID: 18375873). *C. elegans* have a symbiotic relationship with microbes, requiring them for nutrition, optimal development (PMID: 24529378), and drug metabolism (PMID: 23540700). Bacteria can modulate the efficacy of some class of anti-cancer drugs *in vivo*, highlighting that microbial metabolism can impact host response to pharmacotherapy (PMID: 28431245, PMID: 28431244). Therefore, we are not convinced about performing gene expression analysis on dead bacteria taking into consideration the complicated physiological consequences on *C. elegans* by exposing them to dead bacteria even without cisplatin. We are concerned that inducing metabolic shifts in worms by exposure to dead bacteria will affect physiology and health of the worms rather than answer the question about effect of cisplatin on healthy worms. We would like to note that our protocol has always involved adding a concentrated spot of live bacteria onto the cisplatin plates just before the worms are placed on the test plates for precisely this reason. Thus, in our assay worms are exposed to live bacteria so as to not include the complications of the effects of dead bacteria on worm physiology.

We agree that examining the nature of cisplatin induced death in adult animals would be beneficial for this study. We have now examined the effect of the excretory dysfunction (**Fig. S15**) and concluded that indeed mutants with excretory canal defects were cisplatin sensitive. However, adult animals after 18h cisplatin exposure did not display any visible excretory canal defect. We speculate that the lack of a visible excretory canal defect could be because worms live a relatively short period of time after the excretory canals are properly established, whereas excretory canal defective mutants are born with excretory dysfunction. The experiments with worms expressing an excretory canal marker show that there is no large-scale degeneration of these cells. We excluded neurotoxicity as the cause of death in the mutants based on the evidence from the literature that cisplatin treatment did not cause broad damage to sensory functions of *C. elegans*. We have included in the discussion section how innate immune genes protect against cisplatin induced death.

1. We would like to thank the reviewer for his comment on writing and have corrected accordingly.
2. The indicated sentence in the abstract has been corrected by addition of the verb.
3. We have corrected the writing of the manuscript by removing injudicious use of adverbs such as “extremely sensitive” and “very important”.
4. We agree with the reviewer that reporting of LD50 for wild-type animals would be important. We extended the x scale in the **Figure S2** up to 600 $\mu\text{g}/\text{mL}$ and show now that LD50 for 1-day old adult wild-type animals is 550 $\mu\text{g}/\text{mL}$.
5. Indeed the AID system can be used to assess where *sek-1* is required by comparing the effect of knock down of intestinal or hypodermal *sek-1*. We have used the system where 1-day old *sek-1::cMyc::AID (syb2456)* adult animals expressing either the TIR1 intestinal driver *ieSi61* or TIR1 hypodermal driver *reSi1* were exposed to 1mM auxin treatment followed by cisplatin exposure. We found that intestinal knock down of *sek-1* led to a significant decrease in survival upon cisplatin treatment whereas the hypodermal knock down of *sek-1* did not have any effect (**Fig. S9**). This was consistent with our notion that intestine is the place of cisplatin action in adults and rule out a significant role for p38 MAPK pathway activation in the hypodermis upon cisplatin treatment.

REVIEWER #2

Cisplatin is a widely used chemotherapeutic drug for cancer treatment. However, it is known to kill both proliferating cells and post-mitotic cells (such as inner ear hair cells and kidney cells), which rules out its applications in some vulnerable patients due to the damages to these organs. Thus, it is very important to understand the underlying mechanisms by which cisplatin kills non-dividing cells. In this study, Raj et al. used C. elegans as a model to investigate the mechanisms of cisplatin chemoresistance because all somatic cells in adult C. elegans are post-mitotic. Combining C. elegans genetics with proteomic analyses and stress assays, the authors showed that the innate immune response branch downstream of PMK-1/P38 MAPK is critical for the cisplatin chemoresistance. Specifically, they revealed that the transcription factor ATF-7 and multiple downstream immune effectors play essential role in the killing effect of cisplatin on post-mitotic cells. Overall, this study was well conducted, and the findings are important. Below are my remaining critiques:

1) 24 hour cisplatin treatment is sufficient to kill many innate immunity mutant worms. How are these worms killed in general? Some imaging experiments could be done to elaborate the nature of cisplatin-induced cell death. Apoptosis, necrosis, or other forms of cell death? Does cell death only occur to the gut or other tissues as well?

2) For inner ear hair cells, there are studies showing that ROS and detoxification systems are essential for the cisplatin toxicity. But this does not seem to be the case in this study, as SKN-1 and detox system are clearly not required. How to reconcile this discrepancy?

3) In Fig 5a, the survival result of pmk-1(km25) mutant and atf-7(qd22) mutant seems additive. If atf-7 mediates pmk-1's role in cisplatin chemoresistance, should double mutant behave similarly to the atf-7 single mutant? An additive effect would suggest two parallel pathways to me.

4) The P38 MAPK pathway is very conserved across species. Does the current finding apply to mammals? The authors can at least verify their results using mammalian cell culture systems.

REVIEWER #2 RESPONSE:

1) We would like to thank the reviewer for their comments. We agree that experiments elaborating on the nature of cisplatin induced death in 1-day old adult animals would be beneficial for this study. We have now examined apoptosis, necrosis and autophagy using reporter transgenes and include some of the imaging experiments in the manuscript (**Fig. S13, Fig. S14**). There is a high level of necrotic cell death in the sensitive mutants and we find that the necrotic vacuoles have the phosphatidyl serine engulfment signal on the vacuoles. We do not find any significant effect of apoptosis and autophagy in the somatic cells. We conclude that necrosis is the most probable cause of death of the innate immune mutants.

2) We do not exclude the possibility of ROS mediated activation of the pathway based on our finding that cisplatin indeed induced levels of carbonylated proteins (Fig. S1) as a marker for induction of ROS.

In the discussion section we speculate that ROS levels produced as a result of cisplatin exposure were not sufficient to directly produce damage that leads to death. Hourihan *et al.* 2016 (PMID: 27540856) have shown that in the presence of ROS, IRE-1 has a redox related function that is distinct from its UPR^{ER} function and that IRE-1 sulfenylation by ROS is important for this process. Consistent with their findings we also show the role for IRE-1 and TRF-1 in cisplatin response as upstream activators of the p38 MAPK pathway. We assign IRE-1 to a role upstream of the p38MAPK pathway by means of epistasis analysis using gain of function mutants. We also show that the role of IRE-1 is likely not through UPR^{ER} since *xbp-1* mutants are not sensitive and *hsp-4:gfp* is not induced by cisplatin (**Fig. 7**). Furthermore, work from other groups shows ROS induction often leads to induction of inflammatory processes, which stands in agreement with our data showing importance of innate immune response in cisplatin toxicity. Thus the protection against ototoxicity provided by ROS quenchers in inner ear hair cells is not inconsistent with our hypothesis that ROS acts to sulfenylate IRE-1 and quenching ROS would lead to decreased p38MAPK pathway activation. We have now added this point to the discussion section statement on this issue.

3) We agree with the reviewer's comment about the additive effect of cisplatin on the *pmk-1(km25);atf-7(qd22)* double mutants in comparison to the single mutants in *pmk-1(km25)* or *atf-7(qd22)*.

We think that one possible explanation of this finding is the presence of another PMK-1-dependent downstream transcription factor that might modulate cisplatin response in parallel with ATF-7. An alternative explanation is that there is cross-talk between the p38MAPK pathway and the JNK/MAPK pathway so that PMK-1/MAPK has a role in signaling through the JNK pathway (PMID:15256594).

4) We would like to thank the reviewer for this suggestion. We agree that validation of the results in the mammalian system will be valuable. However, while cultured cells have provided insight on tumor development and mechanisms of therapeutic actions, they lack the phenotypic and genetic heterogeneity of the original tumor. Moreover, the cell culture system does not allow us to study the cisplatin response in post-mitotic cells since forcing cultured cells into a post-mitotic state would likely make them unhealthy. An alternative for translating results obtained here to mammalian systems might be using 3D organoids such as kidney organoids and testing them after they have reached maturity and ceased to divide. This would be a long-term project.

REVIEWER #3

Raj and colleagues present a manuscript describing resistance of post-mitotic cells to cisplatin exposure that is dependent on the p38 pathway in C. elegans. Using characterized loss-of-function mutant alleles in p38 pathway components, the authors demonstrate that the signaling cascade of TIR-1/NSY-1/SEK-1/PMK-1 is required for resistance against cisplatin toxicity. This response they show is independent of the downstream transcription factor SKN-1 and instead dependent on the transcription factor ATF-7. By depleting SEK-1 using a somatic tissue specific degron, the authors demonstrate that SEK-1 is required in somatic adult cells for cisplatin resistance. Unbiased proteomics of cisplatin exposed animals revealed many differentially regulated proteins, including innate immune effectors. The authors demonstrate that cisplatin exposure transcriptionally induces three innate immune effector genes in a p38 dependent manner. Additionally, the authors provide data that suggests that cisplatin exposure induces phosphorylation of SEK-1 and ATF-7. Using proteomics, the authors further identified 22 proteins that are induced by cisplatin in a sek-1 dependent manner. Finally, they demonstrate that loss-of-function mutants in three proteins identified in their proteomics analysis are required for cisplatin resistance.

The data presented by the authors that the p38 PMK-1 pathway is required for protection against cisplatin toxicity is convincing. However, I do not think that this manuscript presents enough of a mechanistic advance to be appreciated by a general audience. It is well described in the C. elegans literature that the p38 signaling pathway is required for resistance against a diverse number of stressors (e.g., toxins, infection, heat, and osmotic). Thus, it is not surprising that the p38 pathway is also required for cisplatin resistance. Furthermore, it has been previously documented in the mammalian literature that the p38 pathway is required for cisplatin resistance. It is also important to note that the authors did not demonstrate that cisplatin induces p38 phosphorylation, and instead rely on transcriptional studies of three of seven genes. Also, of note, many of the most significantly regulated proteins in their proteomics experiment are components of the cytosolic and endoplasmic reticulum unfolded protein responses, rather than innate immune effectors.

In my view, two key questions are not addressed in this manuscript, which would considerably elevate the impact of the story. First, how is the p38 pathway activated during cisplatin exposure? Second, how are the genes mobilized by the p38 pathway during cisplatin exposure providing protection against this toxic agent? In the absence of these insights, I do not think that this manuscript rises to a level appropriate for publication in Nature Communications.

Finally, I also have concerns about the central claim of the paper that the innate immune system promotes cisplatin chemoresistance. The authors make this claim based on the fact that individual genes dependent on the p38 pathway in the response to cisplatin have been annotated as “innate immune” genes. In of itself, however, this does not provide a functional connection between innate immunity and cisplatin resistance, as these effectors could have multiple functions in different contexts.

REVIEWER #3 RESPONSE:

We would like to thank the reviewer for finding our data convincing. We agree that the presence of the mechanism by which p38 MAPK pathway is activated upon cisplatin treatment would be of significant interest for a general audience. In the course of our analysis we have established the importance of IRE-1 upstream of the p38 MAPK pathway for the cisplatin response. This response was independent from canonical UPR^{ER} activity of IRE-1 since *xbp-1* mutants were resistant to cisplatin treatment (**Fig. 7**) and we did not see induction of UPR^{ER} marker *hsp-4*.

We agree with the reviewer's comment that the p38MAPK signaling pathway is required for resistance against a diverse number of stressors. However, all of those responses are mainly dependent on the activation of the SKN-1 transcription factor. Our work shows for the first-time the involvement of the innate immune activation via ATF-7 in cisplatin response. Our work also shows that the activation of the type II detoxification genes is not important for protection against cisplatin unlike the role of these detoxification proteins in response to other pro-oxidants like arsenite and paraquat. We note that the role of the ATF-7 transcription factor is specific for the induction of immune effector genes. Moreover, we found that mutants in NSY-1/MAPKKK and the upstream adaptor protein TIR-1 are also cisplatin sensitive. Further we show in terms of timing that SEK-1 and PMK-1 are activated before ATF-7. We find that the IRE-1/TRF-1 module is important for pathway activation and acts upstream of the pathway.

We agree with the reviewer that we did not directly show the activation of the p38 MAPK pathway. Now, instead of relying on the transcriptional studies, we also show that the p38 MAPK pathway is indeed activated by cisplatin in the time dependent manner by western blot analysis using an anti-phospho-p38 antibody (**Fig. S3**). Moreover, we examined the nuclear localization of the PMK-1:mNeonGreen in the intestinal cells and saw an increased number of nuclei with the tagged PMK-1. Nuclear accumulation of PMK-1 occurs after its activation. Thus, indeed we now see the activation of PMK-1 with both these assays.

We also agree with the reviewer that many of the proteins differentially regulated by cisplatin belong to the cytosolic and endoplasmic reticulum unfolded protein responses. However, it does not exclude the possibility of them being also involved in innate immunity response considering their dependence of the presence of SEK-1 revealed by our proteomics analysis. We are aware of the possibility that cisplatin may activate different pathways in parallel since cisplatin is known to have a broad spectrum of action. We have now included this point in the discussion.

In the discussion we have included a section on possible mechanisms via which these gene products mobilized by the p38 pathway provide protection against cisplatin.

Reviewer #3 says that *"The authors make this claim based on the fact that individual genes dependent on the p38 pathway in the response to cisplatin have been annotated as "innate immune" genes. In of itself, however, this does not provide a functional connection between innate immunity and cisplatin resistance, as these effectors could have multiple functions in different contexts."*

Our conclusions on the importance of the innate immune system is not based just on the analysis of individual immune effectors. Rather, our work here shows that the entire canonical immune signaling pathway (TIR-1, NSY-1, SEK-1 and PMK-1) acts in this process. Notably, mutants in *nsy-1*, the *tir-1(qd4)* allele and *atf-7* mutants only have defects in innate immune response. We also show that the cisplatin sensitivity of *tir-1(qd4)* mutants is suppressed by a *nsy-1* gain-of function mutant consistent with its operation downstream of TIR-1. We also show that SEK-1/MAPKK and PMK-1/MAPK are activated before the downstream transcription factor ATF-7. We find in new experiments (by genetic epistasis with loss of function; gain of function double mutant analysis) that the IRE-1/TRF-1 module acts upstream of the p38 MAPK pathway and that hyperactivation of the signaling pathway with gain-

of function mutants can suppress the sensitivity phenotypes of *ire-1* mutants. We also show that *ire-1* mutants diminish the expression of an ATF-7 dependent immune gene reporter. Thus, a robust connection is made at all levels of the pathway. Some of these experiments are new and included in the revised manuscript thanks to the remarks by all three reviewers.

We believe that the original data supplemented with the new experiments provided in the manuscript about involvement of the known immune effectors in the cisplatin response as well as cisplatin sensitivity phenotype of the *C. elegans* immune mutants provides strong evidence for a functional connection between innate immunity and cisplatin resistance. As mentioned before, we do not claim that the innate immune response is the only pathway involved in cisplatin chemoresistance considering the broad spectrum of cisplatin action in the cell. However, we think that our proteomics approach in combination with genetic analysis and biochemical assays provide strong evidence for innate immune response playing a significant role in cisplatin action. As we have discussed, the innate immune system plays a role in protecting tissues and organs against several different insults such as brain injury, wounding or ischaemic stroke.

Finally, we would like to thank the reviewers for raising many interesting and valid points that have improved the manuscript. We hope that our response sheds light on all of the conceptual as well as methodological concerns pointed out by the reviewers. Hopefully we have removed any doubt and that additional experiments provide enough mechanistic insight into how the p38 MAPK pathway is activated during cisplatin exposure.

REVIEWER COMMENTS

Reviewer #1 (Remarks to the Author):

the authors have responded to critiques to my satisfaction.

Reviewer #2 (Remarks to the Author):

Although cisplatin is a widely used chemotherapy agent, its application has been limited by the fact that, in addition to the proliferating tumor cells, it kills many post-mitotic cells in tumors, kidneys, and neurons. Therefore, a mechanistic study on why cisplatin has such a great impact on post-mitotic cells is very important. In this revised manuscript, Raj et al. performed extra experiments and addressed some of my previous concerns. However, two major concerns remain.

1) As the overall impact of this study is on the mechanisms by which cisplatin kills post-mitotic cells and its relevance to the side effects of cisplatin in cancer chemotherapy. Although authors have nicely shown that cisplatin kills *C. elegans* through the canonic p38 pathway, my overall enthusiasm was significantly dampened by the lack of any evidence on whether the same mechanism applies to its side effect in killing post-mitotic cell in mammalian systems.

2) It is true that heat-killed OP50 bacteria lacks vitamin B2 that is required for larval growth. How about UV-killed or streptomycin-killed OP50? Alternatively, authors can switch their worms from live OP50 to dead OP50 after the worms pass their last larval stage, and then conduct their experiments 1-2 days later. It is not uncommon in the *C. elegans* aging research field to use dead OP50 in lifespan experiments. This will address the concern that the activation of p38 pathway is due to cisplatin or live bacteria.

Reviewer #3 (Remarks to the Author):

In my previous review of the manuscript by Raj and colleagues, the major concerns were: (i) lack of mechanistic insight into how cisplatin activates the p38 pathway, (ii) absence of data demonstrating that cisplatin induces p38 phosphorylation, (iii) how the specific p38 dependent immune effectors induced by cisplatin provide resistance against toxicity and (iv) the overall claim of the authors that the innate immune system provides resistance against cisplatin.

In the revised manuscript, the authors provide new data to address only point (ii), while points (i), (iii), and (iv) are inadequately addressed, and without the addition of new data or insights. Thus, I still have major concerns about the suitability of this manuscript for publication.

#1. To address point (i), the authors argue that ROS induced by cisplatin treatment leads to the activation of the p38 pathway through the known IRE-1 – TRF-1 – NSY-1 signaling cascade. Their genetic evidence only partially supports this conclusion. The authors do not perform genetic epistasis experiments with *ire-1* loss-of-function(lf) mutants and loss-of-function mutants in any component of the p38 pathway. Thus, while *ire-1* is required for resistance against cisplatin toxicity, without a genetic epistasis experiment with p38 pathway components it is impossible to assess whether they function in the same or parallel pathways.

#2. Furthermore, the authors demonstrate that *tir-1* is required for resistance against cisplatin toxicity and propose that *tir-1* functions downstream of *ire-1*. I also have a major concern with this conclusion. Other investigators demonstrated in a beautiful and very rigorous study (PMID 27540856) that a sulfenylation-mediated switch causes IRE-1 to activate the p38 PMK-1 pathway through direct activation of NSY-1, and not via TIR-1. In lines 475-485, the authors attempt to reconcile these contradictory findings; however, they did not provide any data to support their conclusions.

Because the authors conclusions challenge an established model, the authors must further characterize how TIR-1 interacts and is activated by IRE-1 during cisplatin exposure, in order to

advance their proposed mechanism. Biochemical or molecular data that could be added to support their hypothesis include: (i) demonstrate that cisplatin induced ROS is required for p38 activation by treating with a ROS scavenger, (ii) demonstrate that cisplatin induces IRE-1 sulfenylation, and (iii) assess the importance of IRE-1 sulfenylated residues for cisplatin resistance by either characterizing previously published strains or by introducing the mutations at the endogenous locus using CRISPR genome editing.

#3. The authors provide new data assessing activation of the p38 pathway during cisplatin exposure by western blotting for phosphorylated p38 and by assessing the nuclear localization of p38. I have concerns with these data. In the western blot experiment, the change in phosphorylation upon cisplatin treatment is not readily apparent and only one western blot is shown with no mention of replicates. The authors should provide quantification of western blot data from at least three biological replicates. The authors used a PMK-1::FLAG strain for total PMK-1. Antibodies for the total PMK-1 protein are readily available and should be used to ensure that total levels of protein are not changing. In the nuclear localization experiment in Fig. S3, there appears to be a high level of background fluorescence and it is not apparent how specific the signal is for fluorescently labeled PMK-1. Is this autofluorescence? The authors should perform pmk-1(RNAi) to assess the specificity of the signal.

#4. I also still have a major problem with the title and abstract of this paper that implicates "The innate immune system." The p38 MAPK study is a stress response pathway. The authors show that this stress response pathway is activated by cisplatin. There is nothing in this paper that ties an "immune response" to the mechanism of cisplatin resistance. Just because ATF-7 controls this response or that there are genes with a GO annotation of "immune genes" does not make the cisplatin response an "immune response." Rather, this is a stress response to cisplatin toxicity that is controlled by the p38 PMK-1 pathway and the transcription factor ATF-7. It is my opinion that the words "immune response" should be removed from the title and abstract

Additional major points of concern.

1. In the proteomics experiment it is not clear that the effectors induced by cisplatin are dependent on SEK-1 for their induction. From these data, it appears that SEK-1 regulates many proteins independent of cisplatin exposure. This is consistent with the role of the p38 pathway in regulating basal immune effector induction. The authors should assess specifically how many cisplatin induced proteins are dependent on SEK-1 for their induction and independent of basal regulation. Indeed, the proteins most significantly by cisplatin exposure appear to be effectors that are not dependent on SEK-1.
2. In the qRT-PCR experiment with pmk-1(km25) the authors only demonstrate that 4 out of 7 p38 dependent transcripts are induced by cisplatin. This calls into question whether the p38 pathway is being induced by cisplatin.
3. T24B8.5 is not induced by cisplatin when analyzed by qRT-PCR yet induces the transcriptional reporter T24B8.5p::gfp. The authors should assess this discrepancy.
4. The epistasis experiment with pmk-1 and atf-7 suggest that atf-7 functions either downstream or in a parallel pathway. The authors should characterize these animals further using transcriptomics or proteomics to elucidate whether atf-7 functions downstream or in a parallel pathway.
5. The authors show data indicating that ATF-7 and SEK-1 are phosphorylated upon cisplatin treatment. This conclusion is made from western blot experiments which demonstrate that cisplatin treatment leads to the presence of a band with an increased molecular weight. I have several problems with these data. The specificity of the anti-body for ATF-7::3xFLAG is unclear. In pmk-1(lf);ATF-7::3xFLAG animals there are no bands present. While loss of the potentially phosphorylated band is consistent with PMK-1 activation, the non-phosphorylated band should remain. Thus, it is possible that the non-phosphorylated band, which the authors use to interpret their data, is non-specific. Furthermore, the authors do not provide evidence that the larger band is due to phosphorylation and not another modification. To make this conclusion the authors should phosphatase treat the samples and perform western blot analysis. All western blot experiments should be repeated three times in biological triplicate with densitometric quantification of the blots.
6. The authors only assess gene expression in untreated atf-7(qd22) animals. To make the conclusion that atf-7 is required for the induction stress responses during cisplatin exposure, the

authors should assess the transcriptional responses in these animals with and without cisplatin treatment. –

7. The authors identify 4 effectors that are induced by and mediate resistance against cisplatin toxicity. In the data presented, it is not clear whether these effectors are dependent on p38 pathway signaling. The authors should perform genetic epistasis experiments with *pmk-1(lf)* animals to further support this conclusion.

Manuscript Title: **The innate immune system promotes cisplatin chemoresistance in post-mitotic *C. elegans* via activation of the p38/MAPK pathway**

REVIEWER #2

Although cisplatin is a widely used chemotherapy agent, its application has been limited by the fact that, in addition to the proliferating tumor cells, it kills many post-mitotic cells in tumors, kidneys, and neurons. Therefore, a mechanistic study on why cisplatin has such a great impact on post-mitotic cells is very important. In this revised manuscript, Raj et al. performed extra experiments and addressed some of my previous concerns. However, two major concerns remain.

*1) As the overall impact of this study is on the mechanisms by which cisplatin kills post-mitotic cells and its relevance to the side effects of cisplatin in cancer chemotherapy. Although authors have nicely shown that cisplatin kills *C. elegans* through the canonic p38 pathway, my overall enthusiasm was significantly dampened by the lack of any evidence on whether the same mechanism applies to its side effect in killing post-mitotic cell in mammalian systems.*

*2) It is true that heat-killed OP50 bacteria lacks vitamin B2 that is required for larval growth. How about UV-killed or streptomycin-killed OP50? Alternatively, authors can switch their worms from live OP50 to dead OP50 after the worms pass their last larval stage, and then conduct their experiments 1-2 days later. It is not uncommon in the *C. elegans* aging research field to use dead OP50 in lifespan experiments. This will address the concern that the activation of p38 pathway is due to cisplatin or live bacteria.*

REVIEWER #2 RESPONSE:

1) We agree with the comment that although the validation of the results in the mammalian system would be very valuable, the idea behind the use of *C. elegans* as a post-mitotic model in this project was the simplicity of the model in comparison to the mammalian system. As we mentioned previously, the study of the post-mitotic cells in the mammalian system is not easy since cultured mammalian cells are largely proliferative hence getting a sufficient number of cells to look at the post-mitotic roles of cisplatin is difficult. Arresting the proliferation of such cells by pharmacological or genetic means represents to our mind an artificial post-mitotic setup which will have its own problems. Adult *C. elegans* provides a unique genetic model in that all somatic tissues are post-mitotic and hence can be used as a model for this action of cisplatin. We note that although in *C. elegans* the germline is proliferative, we have shown by targeted SEK-1 depletion that the effect is not via germline. Lastly, our expertise is in *C. elegans* biology and the contribution of work in worms cellular stress and signaling pathways is clear and this journal has a long record in highlighting contributions of analysis using *C. elegans* to this field.

2) We would like to thank the reviewer for raising that valid point. In order to address the concern about live bacteria leading to activation of the p38 MAPK pathway, we have performed the analysis of immune effector genes mRNA levels in the worms that were exposed to cisplatin without the presence of any bacteria. Worms were washed extensively in M9 before analysis in order to remove the traces of the live bacteria from the surface of the worms as well as from the intestine of the worms. Worms defecate residual bacteria from the intestine after about 15 minutes. This time period was used for the preparation of worms. Washed worms were exposed to cisplatin solution by rotation allowing for uniform exposure to the drug. mRNA levels were assessed by qPCR. These experiments showed that showed the induction of the *clec-4*, *catp-3*, and *dod-24* genes by exposure to cisplatin solution in the absence of bacteria was robust and cisplatin-dependent (**Figure S15**). Further, the same exposure of worms to cisplatin solution caused the induction of the known *pmk-1*-dependent immune genes (**Figure 4c**). Moreover, exposure to cisplatin in the absence of the bacteria caused the induction of the p38 MAPK pathway assessed by the presence of the phosphorylated p38 protein (**Figure 1a**).

We hope that those experiments remove any concern about the p38 MAPK pathway being activated by cisplatin and not by the effect of live bacteria.

REVIEWER #3

In my previous review of the manuscript by Raj and colleagues, the major concerns were: (i) lack of mechanistic insight into how cisplatin activates the p38 pathway, (ii) absence of data demonstrating that cisplatin induces p38 phosphorylation, (iii) how the specific p38 dependent immune effectors induced by cisplatin provide resistance against toxicity and (iv) the overall claim of the authors that the innate immune system provides resistance against cisplatin.

In the revised manuscript, the authors provide new data to address only point (ii), while points (i), (iii), and (iv) are inadequately addressed, and without the addition of new data or insights. Thus, I still have major concerns about the suitability of this manuscript for publication.

#1. To address point (i), the authors argue that ROS induced by cisplatin treatment leads to the activation of the p38 pathway through the known IRE-1 – TRF-1 – NSY-1 signaling cascade. Their genetic evidence only partially supports this conclusion. The authors do not perform genetic epistasis experiments with ire-1 loss-of-function(lf) mutants and loss-of-function mutants in any component of the p38 pathway. Thus, while ire-1 is required for resistance against cisplatin toxicity, without a genetic epistasis experiment with p38 pathway components it is impossible to assess whether they function in the same or parallel pathways.

#2. Furthermore, the authors demonstrate that tir-1 is required for resistance against cisplatin toxicity and propose that tir-1 functions downstream of ire-1. I also have a major concern with this conclusion. Other investigators demonstrated in a beautiful and very rigorous study (PMID 27540856) that a sulfenylation-mediated switch causes IRE-1 to activate the p38 PMK-1 pathway through direct activation of NSY-1, and not via TIR-1. In lines 475-485, the authors attempt to reconcile these contradictory findings; however, they did not provide any data to support their conclusions. Because the authors conclusions challenge an established model, the authors must further characterize how TIR-1 interacts and is activated by IRE-1 during cisplatin exposure, in order to advance their proposed mechanism. Biochemical or molecular data that could be added to support their hypothesis include: (i) demonstrate that cisplatin induced ROS is required for p38 activation by treating with a ROS scavenger, (ii) demonstrate that cisplatin induces IRE-1 sulfenylation, and (iii) assess the importance of IRE-1 sulfenylated residues for cisplatin resistance by either characterizing previously published strains or by introducing the mutations at the endogenous locus using CRISPR genome editing.

#3. The authors provide new data assessing activation of the p38 pathway during cisplatin exposure by western blotting for phosphorylated p38 and by assessing the nuclear localization of p38. I have concerns with these data. In the western blot experiment, the change in phosphorylation upon cisplatin treatment is not readily apparent and only one western blot is shown with no mention of replicates. The authors should provide quantification of western blot data from at least three biological replicates. The authors used a PMK-1::FLAG strain for total PMK-1. Antibodies for the total PMK-1 protein are readily available and should be used to ensure that total levels of protein are not changing. In the nuclear localization experiment in Fig. S3, there appears to be a high level of background fluorescence and it is not apparent how specific the signal is for fluorescently labeled PMK-1. Is this autofluorescence? The authors should perform pmk-1(RNAi) to assess the specificity of the signal.

#4. I also still have a major problem with the title and abstract of this paper that implicates “The innate immune system.” The p38 MAPK study is a stress response pathway. The authors show that this stress response pathway is activated by cisplatin. There is nothing in this paper that ties an “immune response” to the mechanism of cisplatin resistance. Just because ATF-7 controls this response or that there are genes with a GO annotation of “immune genes” does not make the cisplatin response an “immune response.” Rather, this is a stress response to cisplatin toxicity that is controlled by the p38 PMK-1 pathway and the transcription factor ATF-7. It is my opinion that the words “immune response” should be removed from the title and abstract.

Additional major points of concern.

1. *In the proteomics experiment it is not clear that the effectors induced by cisplatin are dependent on SEK-1 for their induction. From these data, it appears that SEK-1 regulates many proteins independent of cisplatin exposure. This is consistent with the role of the p38 pathway in regulating basal immune effector induction. The authors should assess specifically how many cisplatin induced proteins are dependent on SEK-1 for their induction and independent of basal regulation. Indeed, the proteins most significantly by cisplatin exposure appear to be effectors that are not dependent on SEK-1.*
2. *In the qRT-PCR experiment with pmk-1(km25) the authors only demonstrate that 4 out of 7 p38 dependent transcripts are induced by cisplatin. This calls into question whether the p38 pathway is being induced by cisplatin.*
3. *T24B8.5 is not induced by cisplatin when analyzed by qRT-PCR yet induces the transcriptional reporter T24B8.5p::gfp. The authors should assess this discrepancy.*
4. *The epistasis experiment with pmk-1 and atf-7 suggest that atf-7 functions either downstream or in a parallel pathway. The authors should characterize these animals further using transcriptomics or proteomics to elucidate whether atf-7 functions downstream or in a parallel pathway.*
5. *The authors show data indicating that ATF-7 and SEK-1 are phosphorylated upon cisplatin treatment. This conclusion is made from western blot experiments which demonstrate that cisplatin treatment leads to the presence of a band with an increased molecular weight. I have several problems with these data. The specificity of the anti-body for ATF-7::3xFLAG is unclear. In pmk-1(lf);ATF-7::3xFLAG animals there are no bands present. While loss of the potentially phosphorylated band is consistent with PMK-1 activation, the non-phosphorylated band should remain. Thus, it is possible that the non-phosphorylated band, which the authors use to interpret their data, is non-specific. Furthermore, the authors do not provide evidence that the larger band is due to phosphorylation and not another modification. To make this conclusion the authors should phosphatase treat the samples and perform western blot analysis. All western blot experiments should be repeated three times in biological triplicate with densitometric quantification of the blots.*
6. *The authors only assess gene expression in untreated atf-7(qd22) animals. To make the conclusion that atf-7 is required for the induction stress responses during cisplatin exposure, the authors should assess the transcriptional responses in these animals with and without cisplatin treatment.*
7. *The authors identify 4 effectors that are induced by and mediate resistance against cisplatin toxicity. In the data presented, it is not clear whether these effectors are dependent on p38 pathway signaling. The authors should perform genetic epistasis experiments with pmk-1(lf) animals to further support this conclusion.*

REVIEWER #3 RESPONSE:

#1. The genetic epistasis analysis between *ire-1(lf)* mutants and loss-of-function mutants in any component of the p38 pathway is difficult experimentally to perform because of the differences in cisplatin sensitivity profiles of the different mutants. We did make the double mutants between *pmk-1(lf)* and *ire-1(lf)* but the wide range in cisplatin sensitivities made it difficult to do the analysis. Thus, it was not possible to find the cisplatin concentration for this analysis where a synergistic enhancement (or not) of sensitivity might be seen. To address the issue of whether IRE-1 acts upstream of the pathway or in parallel we have performed qPCR analysis showing that the expression *clec-4*, *lys-1*, *dod-24*, or *catp-3* genes is not induced in *ire-1(lf)* mutants upon cisplatin exposure (**Figure S14**) further proving that IRE-1 and PMK-1/p38 act in the same and not parallel pathway. If IRE-1 were acting in a parallel pathway then there would not be any effect on the induction of these four genes whose expression depends on p38/PMK-1 activity (**Figure S14**). We note that we have also shown in the last version (and included here as well) that T24B8.5::GFP induction is blunted in *ire-1* mutants (**Figure 7h**). This is also consistent with the conclusion that IRE-1 acts upstream of the p38 pathway and not in parallel.

#2. (i) As requested by the reviewer we have added the data showing that cisplatin-induced ROS is required for p38 activation by treating with a ROS scavenger (antioxidant MitoTempo) (**Figure 1c, 1d**). MitoTempo was used in the Hourihan *et al.* (2016) paper (PMID: 27540856) to show the diminished expression of IRE-1/p38/SKN-1 response markers. (ii) We have now demonstrated that cisplatin significantly induces IRE-1 sulfenylation (**Figures 7d and 7e**), and (iii) assessed as requested the importance of IRE-1 sulfenylated residue C663 identified by Hourihan *et al.* 2016 (PMID: 27540856) for cisplatin resistance by introducing this mutation at the endogenous locus (C663S) using CRISPR genome editing (**Figure 7d and 7e**). We find that no sulfenylation is detected in IRE-1^{C663S} mutants.

#3. As requested by the reviewer, we have performed the western blots analysis of the p38/PMK-1 phosphorylation in the wild-type *C. elegans* background instead of the analysis in the PMK-1::3xFLAG background. This analysis showed clearly a statistically significant activation of the p38 MAPK pathway after a 2h cisplatin treatment (**Figures 1a and 1b**). The quantification was based on three independent biological replicates. Antibody for total PMK-1 level (using antibodies obtained from the Pukkila-Worley lab) was used for total levels of PMK-1 (**Figure 1a**). In order to avoid high levels of background fluorescent in animals expressing PMK-1::mNeonGreen, we have used the strain GH403 (*glo-3(kx94)*) which partially lacks gut granules (birefringence) in intestinal cells (**Figure S3a**). The *pmk-1::mNeonGreen; glo-3(kx94)* double mutant was analyzed by fluorescence microscopy (**Figure S3b**). There was significantly reduced intestinal background fluorescence which in turn increased the detection of the PMK-1::mNeonGreen signal. The analysis in the *glo-3* mutants confirms that cisplatin exposure leads to nuclear accumulation of PMK-1::mNeonGreen. We also note that depletion of GLO-3 has been in other studies to overcome the effect of intestinal autofluorescence (for example PMID: 35098926).

#4. We would like to thank the reviewer for that comment. We agree that the title of the manuscript should be improved. Therefore, we propose a new title:

Cisplatin toxicity is counteracted by the activation of the p38/ATF-7 signaling pathway in post-mitotic C. elegans

1. We would like to thank the reviewer for that comment. As noticed by the reviewer, 129 proteins showed decreased abundance with a fold change less than -1.5 in *sek-1* mutants (**Figure S12**) showing that indeed SEK-1 has a broad function in *C. elegans*. This was not surprising considering that SEK-1 regulates many proteins independent of cisplatin exposure and the p38 MAPK pathway is not the only

target for cisplatin. This shows the broad effect of the drug treatment. This notion is further supported by Reactome pathway analysis where many other pathways seem to be affected by cisplatin like vesicle-mediated transport, pathways involved in protein metabolism, or signal transduction (**Table S2**). Indeed, many highly cisplatin-induced proteins were not dependent on the presence of *sek-1*. From 325 proteins that fold change was higher than 1.5 upon cisplatin exposure, only 26 proteins ($FC < -1.5$) seem to be dependent on the presence of SEK-1 (**Table S3**). However, 22 proteins had a decreased abundance with a fold change of less than -1.5 in *sek-1* mutants treated with cisplatin in comparison to the wild-type samples treated with cisplatin. That shows a big dependence of those SEK-1-regulated genes for cisplatin exposure.

As requested by the reviewer we have analyzed the data further and showed that only 6 cisplatin-induced proteins are dependent on SEK-1 for their induction and independent of basal regulation (Figure R1; class F).

Figure R1. Venn diagram of different protein classes regulated by SEK-1 and/or cisplatin. Cisplatin-induced class A proteins ($FDR < 0.05$; $FC > 1.5$). SEK-1 negatively regulated class B proteins ($FDR < 0.05$; $FC < -1.5$) and positively regulated class C proteins ($FDR < 0.05$; $FC > 1.5$). SEK-1 was not required for basal regulation of class D proteins. SEK-1 was required for basal and inducible levels of class E proteins. SEK-1 was not required for basal but inducible levels of class F proteins. Numbers in the brackets represent the number of proteins in each group.

2. The qPCR data presented in **Figure 4a** was initially performed to confirm the innate immune response driven by the p38 MAPK pathway as important for cisplatin resistance. Genes in **Figure 4a** were chosen based on the available literature as known PMK-1-dependent targets induced by *Pseudomonas aeruginosa* PA14 infection (PMID: 17096597, 26818074, 20369020). However, infections with different bacterial pathogens will cause the activation of different innate immune response gene sets (PMID: 17875205, 22236697). Therefore, not all of the genes responsible for protection from PA14 will be induced by cisplatin exposure. Taking that into account as well as information that the cisplatin will cause ROS induction and that immune response might be the consequence of the stress response (PMID: 26016853), the proteomics analysis had been done by us to further identify the genes that might be important effectors for cisplatin resistance downstream of the p38 cascade. Those four effectors (*clec-4*, *lys-1*, *dod-24*, *catp-3*), as requested by the reviewers, have been analyzed experimentally further and their importance for cisplatin resistance has been confirmed (**Figures 8c and 8d**) and their expression is PMK-1 dependent (**Figure S14**). Moreover, our new experiments in wild-type animals directly show that indeed p38/PMK-1 is phosphorylated and therefore activated by cisplatin (**Figure 1a**).

3. Indeed the statistical significance was not present in the qPCR analysis even though the increase of T24B8.5 gene expression was clearly seen. We have seen the same pattern of induction of the F49F1.6 gene upon cisplatin treatment in the wild-type strain (**Figure 4a**) even though the analysis with one-way ANOVA did not confirm statistical significance. This can be explained by the error bars (SEM) present in the graph. Their size did not allow for statistical significance to be observed even though the increase was visible. We saw a statistically significant increase in K08D8.5 gene levels upon cisplatin treatment even though the increase was smaller than one seen in the case of T24B8.5 and F49F1.6. genes, which only further confirms that hypothesis. Therefore, we do not believe that there is a discrepancy between the results from qPCR and T24B8.5p::GFP analysis. In in both cases, the increase upon cisplatin treatment was clearly present.

4. ATF-7 is a known PMK-1-dependent transcription factor. Indeed, the epistasis analysis with *pmk-1* and *atf-7* suggests that *atf-7* functions either downstream or in a parallel pathway. However, as requested we have performed further analysis in *atf-7(qd22)* mutants where we show (i) the lack of the T24B8.5p::GFP induction upon cisplatin treatment (**Figure 5c**) in wild-type and *pmk-1(km25)* background, (ii) phosphorylation of ATF-7 upon cisplatin treatment but not in the cisplatin exposed *pmk-1(km25)* mutant background (**Figure 6c**), (iii) or the lack of the PMK-1-dependent innate immune genes induction in the *atf-7(qd22)* mutant background upon cisplatin treatment (**Figure S7**). These results confirm that the ATF-7 is not only involved in the cisplatin response but also is a downstream target of the PMK-1. Therefore, we believe that proteomics or transcriptomics analysis of the *atf-7* mutants is not necessary in order to confirm its involvement in the cisplatin response or its dependence on *pmk-1*.

5. For the detection of the phosphorylated version of the protein we have used SuperSep™ Phos-tag™ Precast Gels (Fuji, 195-17991; <https://labchem-wako.fujifilm.com/us/product/detail/W01W0119-1799.html>). These are widely used gels that specifically detect the presence of the phosphorylated proteins based on the shift in the molecular weight. Moreover, we have included the wild-type sample in the gel where no band was detected confirming the specificity of the antibody. However, we agree with the reviewer that the Phos-tag gels presented in the main **Figure 6** were not presented with the best quality. Therefore, we have included two additional Phos-tag gels with two additional biological replicates showing (i) an increase in the phosphorylated ATF-7 which confirmed the previous results and gave us the possibility to quantify the increase, (ii) the presence of only the non-phosphorylated band in the *pmk-1(km25);ATF-7::3xFlag* strain which also suggests that the phosphorylated band in the ATF-7::3xFlag strain is the correct band, (iii) the absence of the phosphorylated ATF-7 upon treatment of the lysates with lambda protein phosphatase (LPP), which further confirms the specificity of the band and confirms that the higher molecular weight of the band is indeed the consequence of phosphorylation.

6. We have added data showing the lack of induction of immune gene expression in cisplatin-treated *atf-7(qd22)* animals proving that *atf-7* is required for the induction of stress response (**Figure S7**).

7. Because of the differences in the cisplatin sensitivities between the *pmk-1(km25)* mutants and any of the mutants in the four effectors analyzed in this work, the genetic epistasis analysis was not possible experimentally in order to show that those four effectors are indeed dependent on p38 pathway signaling. Thus, as in the case of double mutant between *pmk-1(lf)* and *ire-1(lf)*, it was not possible to find the cisplatin concentration for this analysis where a synergistic enhancement (or not) of sensitivity might be seen.

To demonstrate the dependence of those four effectors on the p38 MAPK signaling, we have performed the qPCR analysis of those effectors in the *pmk-1(km25)* mutants which showed that lack of the PMK-

1/p38 resulted in low levels of the *clec-4*, *lys-1*, *dod-24*, and *catp-3* expression upon cisplatin exposure (**Figure S14**). If the expression of these genes by cisplatin was independent of PMK-1 activity, their expression should still be observed in *pmk-1(km25)* mutants. Since we see the opposite, we conclude that the expression of *clec-4*, *lys-1*, *dod-24* and *catp-3* is dependent on p38/PMK-1 signaling. Moreover, those data stand in agreement with our proteomics analysis, where protein levels of those 4 effectors were significantly reduced in the absence of SEK-1 showing for the first time their dependence on the p38 MAPK cascade. We hope that this analysis resolves the question about the dependence of *clec-4*, *lys-1*, *dod-24*, or *catp-3* on the activation of the p38 pathway.

REVIEWERS' COMMENTS

Reviewer #2 (Remarks to the Author):

I fully understand that the authors may not have the expertise in mammalian cell biology. In fact, growing a large amount of primary post-mitotic mammalian neurons are not that difficult. Due to the lack of verification in a mammalian model, I suggest that the authors should downplay their statement on discovering the mechanisms of cisplatin toxicity/side effect in treating human cancers.

Reviewer #3 (Remarks to the Author):

The data presented in the revision of this manuscript better support the core conclusions of the authors.

Manuscript Title: **Cisplatin toxicity is counteracted by the activation of the p38/ATF-7 signaling pathway in post-mitotic *C. elegans***

Reviewer #2

I fully understand that the authors may not have the expertise in mammalian cell biology. In fact, growing a large amount of primary post-mitotic mammalian neurons are not that difficult. Due to the lack of verification in a mammalian model, I suggest that the authors should downplay their statement on discovering the mechanisms of cisplatin toxicity/side effect in treating human cancers.

Reviewer #2 Reponses:

Thank you for the suggestion. We have downplayed the statements on discovering the mechanisms of cisplatin toxicity/side effect in treating human cancers.